# DAG Learning on the Permutahedron

**Valentina Zantedeschi**
ServiceNow Research
vzantedeschi@gmail.com

**Luca Franceschi**
Amazon Web Services*
franuluc@amazon.de

**Jean Kaddour**
University College London,
Centre for AI
jean.kaddour.20@ucl.ac.uk

**Matt J. Kusner**
University College London,
Centre for AI
m.kusner@ucl.ac.uk

**Vlad Niculae**
Informatics Institute,
University of Amsterdam
v.niculae@uva.nl

## ABSTRACT

We propose a continuous optimization framework for discovering a latent directed acyclic graph (DAG) from observational data. Our approach optimizes over the polytope of permutation vectors, the so-called *Permutahedron*, to learn a topological ordering. Edges can be optimized jointly, or learned conditional on the ordering via a non-differentiable subroutine. Compared to existing continuous optimization approaches our formulation has a number of advantages including: 1. *validity*: optimizes over exact DAGs as opposed to other relaxations optimizing approximate DAGs; 2. *modularity*: accommodates any edge-optimization procedure, edge structural parameterization, and optimization loss; 3. *end-to-end*: either alternately iterates between node-ordering and edge-optimization, or optimizes them jointly. We demonstrate, on real-world data problems in protein-signaling and transcriptional network discovery, that our approach lies on the Pareto frontier of two key metrics, the SID and SHD.

## 1 INTRODUCTION

In many domains, including cell biology (Sachs et al., 2005), finance (Sanford & Moosa, 2012), and genetics (Zhang et al., 2013), the data generating process is thought to be represented by an underlying directed acylic graph (DAG). Many models rely on DAG assumptions, e.g., causal modeling uses DAGs to model distribution shifts, ensure predictor fairness among subpopulations, or learn agents more sample-efficiently (Kaddour et al., 2022). A key question, with implications ranging from better modeling to causal discovery, is how to recover this unknown DAG from observed data alone. While there are methods for identifying the underlying DAG if given additional interventional data (Eberhardt, 2007; Hauser & Bühlmann, 2014; Shanmugam et al., 2015; Kocaoglu et al., 2017; Brouillard et al., 2020; Addanki et al., 2020; Squires et al., 2020; Lippe et al., 2022), it is not always practical or ethical to obtain such data (e.g., if one aims to discover links between dietary choices and deadly diseases).

Learning DAGs from observational data alone is fundamentally difficult for two reasons. (i) *Estimation*: it is possible for different graphs to produce similar observed data, either because the graphs are Markov equivalent (they represent the same set of data distributions) or because not enough samples have been observed to distinguish possible graphs. This riddles the search space with local minima; (ii) *Computation*: DAG discovery is a costly combinatorial optimization problem over an exponentially large solution space and subject to global acyclicity constraints.

To address issue (ii), recent work has proposed continuous relaxations of the DAG learning problem. These allow one to use well-studied continuous optimization procedures to search the space of DAGs given a score function (e.g., the likelihood). While these methods are more efficient than combinatorial methods, the current approaches have one or more of the following downsides: 1. *Invalidity*: existing methods based on penalizing the exponential of the adjacency matrix (Zheng

---

*Work done prior to joining Amazon.

et al., 2018; Yu et al., 2019; Zheng et al., 2020; Ng et al., 2020; Lachapelle et al., 2020; He et al., 2021) are not guaranteed to return a valid DAG in practice (see Ng et al. (2022) for a theoretical analysis), but require post-processing to correct the graph to a DAG. How the learning method and the post-processing method interact with each other is not currently well-understood; 2. *Non-modularity*: continuously relaxing the DAG learning problem is often done to leverage gradient-based optimization (Zheng et al., 2018; Ng et al., 2020; Cundy et al., 2021; Charpentier et al., 2022). This requires all training operations to be differentiable, preventing the use of certain well-studied black-box estimators for learning edge functions; 3. *Error propagation*: methods that break the DAG learning problem into two stages risk propagating errors from one stage to the next (Teyssier & Koller, 2005; Bühlmann et al., 2014; Gao et al., 2020; Reisach et al., 2021; Rolland et al., 2022).

Following the framework of Friedman & Koller (2003), we propose a new differentiable DAG learning procedure based on a decomposition of the problem into: (i) learning a topological ordering (i.e., a total ordering of the variables) and (ii) selecting the best scoring DAG consistent with this ordering. Whereas previous differentiable order-based works (Cundy et al., 2021; Charpentier et al., 2022) implemented step (i) through the usage of permutation matrices[1], we take a more straightforward approach by directly working in the space of vector orderings. Overall, we make the following contributions to score-based methods for DAG learning:

- We propose a novel vector parametrization that associates a single scalar value to each node. This parametrization is $(i)$ intuitive: the higher the score the lower the node is in the order; $(ii)$ stable, as small perturbations in the parameter space result in small perturbations in the DAG space.

- With such parameterization in place, we show how to learn DAG structures end-to-end from observational data, with any choice of edge estimator (we do not require differentiability). To do so, we leverage recent advances in discrete optimization (Niculae et al., 2018; Correia et al., 2020) and derive a novel top-k oracle over permutations, which could be of independent interest.

- We show that DAGs learned with our proposed framework lie on the Pareto front of two key metrics (the SHD and SID) on two real-world tasks and perform favorably on several synthetic tasks.

These contributions allow us to develop a framework that addresses the issues of prior work. Specifically, our approach: 1. Models sparse distributions of DAG topological orderings, ensuring all considered graphs are DAGs (also during training); 2. Separates the learning of topological orderings from the learning of edge functions, but 3. Optimizes them end-to-end, either jointly or alternately iterating between learning ordering and edges.

## 2 RELATED WORK

The work on DAG learning can be largely categorized into four families of approaches: (a) combinatorial methods, (b) continuous relaxation, (c) two-stage, (d) differentiable, order-based.

**Combinatorial methods.** These methods are either constraint-based, relying on conditional independence tests for selecting the sets of parents (Spirtes et al., 2000), or score-based, evaluating how well possible candidates fit the data (Geiger & Heckerman, 1994) (see Kitson et al. (2021) for a survey). Constraint-based methods, while elegant, require conditional independence testing, which is known to be a hard statistical problem Shah & Peters (2020). For this reason, we focus our attention in this paper on score-based methods. Of these, exact combinatorial algorithms exist only for small number of nodes $d$ (Singh & Moore, 2005; Xiang & Kim, 2013; Cussens, 2011), because the space of DAGs grows superexponentially in $d$ and finding the optimal solution is NP-hard to solve (Chickering, 1995). Approximate methods (Scanagatta et al., 2015; Aragam & Zhou, 2015; Ramsey et al., 2017) rely on global or local search heuristics in order to scale to problems with thousands of nodes.

**Continuous relaxation.** To address the complexity of the combinatorial search, more recent methods have proposed exact characterizations of DAGs that allow one to tackle the problem by continuous optimization (Zheng et al., 2018; Yu et al., 2019; Zheng et al., 2020; Ng et al., 2020; Lachapelle et al., 2020; He et al., 2021). To do so, the constraint on acyclicity is expressed as a smooth function (Zheng et al., 2018; Yu et al., 2019) and then used as penalization term to allow efficient optimization.

---

[1]Critically, the usage of permutation matrices allows to maintain a fully differentiable path from loss to parameters (of the permutation matrices) via Sinkhorn iterations or other (inexact) relaxation methods.

However, this procedure no longer guarantees the absence of cycles at any stage of training, and solutions often require post-processing. Concurrently to this work, Bello et al. (2022) introduce a log-determinant characterization and an optimization procedure that is guaranteed to return a DAG at convergence. In practice, this relies on thresholding for reducing false positives in edge prediction.

**Two-stage.** The third prominent line of works learns DAGs in two-stages: (i) finding an ordering of the variables, and (ii) selecting the best scoring graph among (or marginalizing over) the structures that are consistent with the found ordering (Teyssier & Koller, 2005; Bühlmann et al., 2014; Gao et al., 2020; Reisach et al., 2021; Rolland et al., 2022). As such, these approaches work over exact DAGs, instead of relaxations. Additionally, they work on the space of orderings which is smaller and more regular than the space of DAGs (Friedman & Koller, 2003; Teyssier & Koller, 2005), while guaranteeing acyclicity. The downside of these approaches is that errors in the first stage can propagate to the second stage that is unaware of them.

**Differentiable, order-based.** The final line of works uses the two-stage decomposition above, but addresses the issue of error propagation by formulating an end-to-end differentiable optimization approach (Friedman & Koller, 2003; Cundy et al., 2021; Charpentier et al., 2022). In particular, Cundy et al. (2021) optimizes node ordering using the polytope of permutation matrices (the Birkhoff polytope) via the Gumbel-Sinkhorn approximation (Mena et al., 2018). This method requires $\mathcal{O}(d^3)$ time and $\mathcal{O}(d^2)$ memory complexities. To lower the time complexity, Charpentier et al. (2022) suggest to leverage another operator (SoftSort, Prillo & Eisenschlos, 2020) that drops a constraint on the permutation matrix (allowing row-stochastic matrices). Both methods introduce a mismatch between forward (based on the hard permutation) and backward (based on soft permutations) passes. Further, they require all downstream operations to be differentiable, including the edge estimator.

## 3   SETUP

### 3.1   THE PROBLEM

Let $\mathbf{X} \in \mathbb{R}^{n \times d}$ be a matrix of $n$ observed data inputs generated by an unknown Structural Equation Model (SEM) (Pearl, 2000). An SEM describes the functional relationships between $d$ features as edges between nodes in a DAG $\mathcal{G} \in \mathbb{D}[d]$ (where $\mathbb{D}[d]$ is the space of all DAGs with $d$ nodes). Each feature $\mathbf{x}_j \in \mathbb{R}^n$ is generated by some (unknown) function $f_j$ of its (unknown) parents $\mathrm{pa}(j)$ as: $\mathbf{x}_j = f_j(\mathbf{x}_{\mathrm{pa}(j)})$. We keep track of whether an edge exists in the graph $\mathcal{G}$ using an adjacency matrix $\mathbf{A} \in \{0,1\}^{d \times d}$ (i.e., $\mathbf{A}_{ij} = 1$ if and only if there is a (directed) edge $i \to j$). For example, a special case is when the structural equations are linear with Gaussian noise,

$$\mathbf{y}_j = f_j(\mathbf{X}, \mathbf{A}_j) = \mathbf{X}(\mathbf{w}_j \circ \mathbf{A}_j) + \varepsilon; \quad \varepsilon \sim \mathcal{N}(0, \nu) \tag{1}$$

where $\mathbf{w}_j \in \mathbb{R}^d$, $\mathbf{A}_j$ is the $j$th column of $\mathbf{A}$, and $\nu$ is the noise variance. This is just to add intuition; our framework is compatible with non-linear, non-Gaussian structural equations.

### 3.2   OBJECTIVE

Given $\mathbf{X}$, our goal is to recover the unknown DAG that generated the observations. To do so we must learn (a) the *connectivity parameters* of the graph, represented by the adjacency matrix $\mathbf{A}$, and (b) the *functional parameters* $\boldsymbol{\Phi} = \{\phi_j\}_{j=1}^d$ that define the edge functions $\{f^{\phi_j}\}_{j=1}^d$. Score-based methods (Kitson et al., 2021) learn these parameters via a constrained non-linear mixed-integer optimization problem

$$\min_{\substack{\mathbf{A} \in \mathbb{D}[d] \\ \boldsymbol{\Phi}}} \sum_{j=1}^d \ell\left(\mathbf{x}_j, f^{\phi_j}(\mathbf{X} \circ \mathbf{A}_j)\right) + \lambda \Omega(\boldsymbol{\Phi}), \tag{2}$$

where $\ell : \mathbb{R}^n \times \mathbb{R}^n \to \mathbb{R}$ is a loss that describes how well each feature $\mathbf{x}_j$ is predicted by $f^{\phi_j}$. As in eq. (1), the adjacency matrix defines the parents of each feature $\mathbf{x}_j$ as follows $pa(j) = \{i \subseteq [d] \setminus j \mid \mathbf{A}_{ij} = 1\}$. Only these features will be used by $f^{\phi_j}$ to predict $\mathbf{x}_j$, via $\mathbf{X} \circ \mathbf{A}_j$. The constraint $\mathbf{A} \in \mathbb{D}[d]$ enforces that the connectivity parameters $\mathbf{A}$ describes a valid DAG. Finally, $\Omega(\boldsymbol{\Phi})$ is a regularization term encouraging sparseness. So long as this regularizer takes the same value for all DAGs within a Markov equivalence class, the consistency results of Brouillard et al. (2020, Theorem 1) prove that the solution to Problem (2) is Markov equivalent to the true DAG, given standard assumptions.

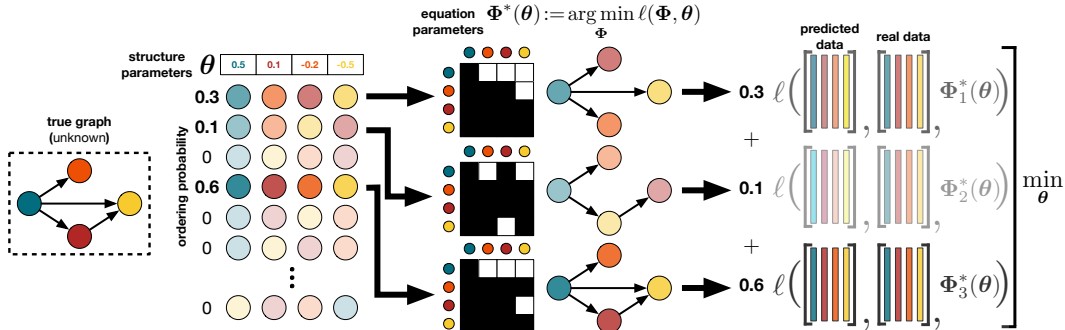

Figure 1: **DAGuerreotype**: Our end-to-end approach to DAG learning works by (a) learning a sparse distribution over node orderings via structure parameters $\theta$, and (b) learning a sparse predictor $\mathbf{w}^*$ to estimate the data. Any black-box predictor can be used to learn $\mathbf{w}^*$; differentiability is not necessary.

### 3.3 SPARSE RELAXATION METHODS

In developing our method, we will leverage recent works in sparse relaxation. In particular, we will make use of the SparseMAP (Niculae et al., 2018) and Top-$k$ SparseMAX (Correia et al., 2020) operators, which we briefly describe below. At a high level, the goal of both these approaches is to relax structured problems of the form $\boldsymbol{\alpha}^\star := \arg\max_{\boldsymbol{\alpha} \in \triangle^D} \mathbf{s}^\top \boldsymbol{\alpha}$ so that $\partial \boldsymbol{\alpha}^\star / \partial \mathbf{s}$ is well-defined (where $\triangle^D := \{\boldsymbol{\alpha} \in \mathbb{R}^D \mid \boldsymbol{\alpha} \succeq 0, \sum_{i=1}^D \alpha_i = 1\}$ is the $D$-dimensional simplex). This will allow $\mathbf{s}$ to be learned by gradient-based methods. Note that both approaches require querying an oracle that finds the best scoring structures. We are unaware of such an oracle for DAG learning, i.e. for $\mathbb{D}[d]$ being the vertices of $\triangle^D$. However, we will show that by decomposing the DAG learning problem, we can find an oracle for the decomposed subproblem. We will derive this oracle in Section 4 and prove its correctness.

**Top-$k$ sparsemax (Correia et al., 2020).** This approach works by (i) regularizing $\boldsymbol{\alpha}$, and (ii) constraining the number of non-zero entries of $\boldsymbol{\alpha}$ to be at most as follows $k$: $\arg\max_{\boldsymbol{\alpha} \in \triangle^D, \|\boldsymbol{\alpha}\|_0 \leq k} \mathbf{s}^\top \boldsymbol{\alpha} - \|\boldsymbol{\alpha}\|_2^2$. To solve this optimization problem, top-$k$ sparsemax requires an oracle that returns the $k$ structures with the highest scores $\mathbf{s}^\top \boldsymbol{\alpha}$.

**SparseMAP (Niculae et al., 2018).** Assume $\mathbf{s}$ has a low-dimensional parametrization $\mathbf{s} = \mathbf{B}^\top \mathbf{r}$, where $\mathbf{B} \in \mathbb{R}^{q \times D}$ and $q \ll D$. SparseMAP relaxes $\boldsymbol{\alpha}^\star = \arg\max_{\boldsymbol{\alpha} \in \triangle^D} \mathbf{r}^\top \mathbf{B} \boldsymbol{\alpha}$ by regularizing the lower-dimensional 'marginal space' $\arg\max_{\boldsymbol{\alpha} \in \triangle^D} \mathbf{r}^\top \mathbf{B} \boldsymbol{\alpha} - \|\mathbf{B} \boldsymbol{\alpha}\|_2^2$. The relaxed problem can be solved using the active set algorithm (Nocedal & Wright, 1999), which iteratively queries an oracle for finding the best scoring structure $(\mathbf{r} - \mathbf{B}^\top \boldsymbol{\alpha}^{(t)})^\top \mathbf{B} \boldsymbol{\alpha}$ at iteration $t + 1$.

## 4 DAG LEARNING VIA SPARSE RELAXATIONS

A key difficulty when learning DAG structures is that the characterization of the set of all valid DAGs: as soon as some edges are added, other edges are prohibited. However, note the following key observation: any DAG can be decomposed as follows (i) Assign to each of the $d$ nodes a rank and reorder nodes according to this rank (this is called a *topological ordering*); (ii) Only allow edges from lower nodes in the order to higher nodes, i.e., from a node $i$ to a node $j$ if $\mathbf{x}_i \prec \mathbf{x}_j$. This approach lies at the core of our method for DAG learning, which we dub *DAGuerreotype*, shown in Figure 1. In this section we derive the framework, present a global sensitivity result, show how to learn both the structural and the edge equations parameters leveraging the sparse relaxation methods introduced above and study the computational complexity of our method.

### 4.1 LEARNING ON THE PERMUTAHEDRON

Given $d$ nodes, let $\Sigma_d$ be the set of all permutations of node indices $\{1, \ldots, d\}$. Given a vector $\mathbf{v} \in \mathbb{R}^d$, let $\mathbf{v}^\sigma := [\mathbf{v}^{\sigma(1)}, \ldots, \mathbf{v}^{\sigma(d)}]^\top$ be the vector of reordered $\mathbf{v}$ according to the permutation $\sigma \in \Sigma_d$. Similarly, for a matrix $\mathbf{M} \in \mathbb{R}$, let $\mathbf{M}^\sigma$ be the matrix obtained by permuting the rows and columns of $\mathbf{M}$ by $\sigma$.

Let $\mathbb{D}_C[d]$ be the set of *complete* DAGs (i.e., DAGs with all possible edges). Let $\mathbf{R} \in \{0, 1\}^{d \times d}$ be the binary strictly upper triangular matrix where the upper triangle is all equal to $1$. Then $\mathbb{D}_C[d]$ can be fully enumerated given $\mathbf{R}$ and $\Sigma_d$, as follows:

$$\mathbb{D}_C[d] = \{\mathbf{R}^\sigma : \sigma \in \Sigma_d, \mathbf{R} \in \{0, 1\}^{d \times d}, \mathbf{R}_{ij} = 0 \ \forall i \geq j, \mathbf{R}_{ji} = 1 \ \forall j < i\}. \tag{3}$$

Therefore it is sufficient to learn $\sigma$ in step (i), and then learn which edges to drop in step (ii).

**The vector parameterization.** Imagine now that $\boldsymbol{\theta} \in \mathbb{R}^d$ defines a score for each node, and these scores induce an ordering $\sigma(\boldsymbol{\theta})$: the smaller the score, the earlier the node should be in the ordering. Formally, the following optimization problem finds such an ordering:

$$\sigma(\boldsymbol{\theta}) \in \arg\max_{\sigma \in \Sigma_d} \boldsymbol{\theta}^\top \boldsymbol{\rho}^\sigma, \quad \text{where } \boldsymbol{\rho} = [1, 2, \ldots, d]. \tag{4}$$

Note that a simple oracle solves this optimization problem: sort $\boldsymbol{\theta}$ into increasing order (as given by The Rearrangement Inequality (Hardy et al., 1952, Thms. 368–369)). We emphasize that we write '$\in$' in eq. (4) since the r.h.s can be a set: in fact, this happens exactly when some components of $\boldsymbol{\theta}$ are equal. Beside being intuitive and efficient, the parameterization of $\mathbb{D}_C[d]$ given by $\boldsymbol{\theta} \mapsto \boldsymbol{R}^{\sigma(\boldsymbol{\theta})}$ allows us to upper bound the structural Hamming distance (SHD) between any two complete DAGs ($\boldsymbol{R}^{\sigma(\boldsymbol{\theta})}$ and $\boldsymbol{R}^{\sigma(\boldsymbol{\theta}')}$) by the number of hyper-planes of "equal coordinates" ($H_{i,j} = \{\boldsymbol{x} \in \mathbb{R}^d : \boldsymbol{x}_i = \boldsymbol{x}_j\}$) that are traversed by the segment connecting $\boldsymbol{\theta}$ and $\boldsymbol{\theta}'$ (see Figure 4 in the Appendix for a schematic). More formally, we state the following theorem.

**Theorem 4.1** (Global sensitivity). *For any $\boldsymbol{\theta} \in \mathbb{R}^d$ and $\boldsymbol{\theta}' \in \mathbb{R}^d$*

$$\text{SHD}\left(\boldsymbol{R}^{\sigma(\boldsymbol{\theta})}, \boldsymbol{R}^{\sigma(\boldsymbol{\theta}')}\right) \leq \int_{t \in [0,1]} \sum_i \sum_{j > i} \delta_{H_{i,j}}(\boldsymbol{\theta} + t(\boldsymbol{\theta}' - \boldsymbol{\theta})) \, \mathrm{d}t \tag{5}$$

*where $\delta_A(x)$ is the (generalized) Dirac delta that evaluates to infinity if $x \in A$ and $0$ otherwise.*

In particular, Theorem 5 shows that we can expect that small changes in $\boldsymbol{\theta}$ (e.g. due to gradient-based iterative optimization) lead to small changes in the complete DAG space, offering a result that is reminiscent to Lipschitz-smoothness for smooth optimization. We defer proof and further commentary (also compared to the parameterization based on permutation matrices) to Appendix C.

**Learning $\boldsymbol{\theta}$ with gradients.** Notice that we cannot take gradients through eq. (4) because (a) $\sigma(\boldsymbol{\theta})$ is not even a function (due to it possibly being a set), and (b) even if we restrict the parameter space not to have ties, the mapping is piece-wise constant and uninformative for gradient-based learning. To circumvent these issues, Blondel et al. (2020) propose to relax problem (4) by optimizing over the convex hull of all permutations of $\boldsymbol{\rho}$, that is the order-$d$ *Permutahedron* $\mathbb{P}[d] := \text{conv}\{\boldsymbol{\rho}^\sigma \mid \sigma \in \Sigma_d\}$, and adding a convex regularizer. These alterations yield to a class of differentiable mappings (soft permutations), indexed by $\tau \in \mathbb{R}^+$,

$$\boldsymbol{\mu}(\boldsymbol{\theta}) = \arg\max_{\boldsymbol{\mu} \in \mathbb{P}[d]} \boldsymbol{\theta}^\top \boldsymbol{\mu} - \frac{\tau}{2} \|\boldsymbol{\mu}\|_2^2, \tag{6}$$

which, in absence of ties, are exact for $\tau \to 0$. This technique is, however, unsuitable for our case, as the $\boldsymbol{\mu}(\boldsymbol{\theta})$'s do not describe valid permutations except when taking values on vertices of $\mathbb{P}[d]$. Instead, we show next how to obtain meaningful gradients whilst maintaining validity adapting the sparseMAP or the top-$k$ sparsemax operators to our setting.

**Leveraging sparse relaxation methods.** Let $D = d!$ be the total number permutations of $d$ elements, and $\triangle^D$ be the $D$-dimensional simplex. We can (non-uniquely) decompose $\boldsymbol{\mu} = \sum_{\sigma \in \Sigma_d} \boldsymbol{\alpha}_\sigma \boldsymbol{\rho}^\sigma$ for some $\boldsymbol{\alpha} \in \triangle^D$. Plugging this into eq. (6) leads to

$$\boldsymbol{\alpha}^{\text{sparseMAP}}(\boldsymbol{\theta}) \in \arg\max_{\boldsymbol{\alpha} \in \triangle^D} \boldsymbol{\theta}^\top \mathbb{E}_{\sigma \sim \boldsymbol{\alpha}}[\boldsymbol{\rho}_\sigma] - \frac{\tau}{2} \|\mathbb{E}_{\sigma \sim \boldsymbol{\alpha}}[\boldsymbol{\rho}_\sigma]\|_2^2, \tag{7}$$

We can recognize in (7) an instance of the SparseMAP operator introduced in Section 3.3. Among all possible decomposition, we will favor sparse ones. This is achieved by employing the active set algorithm (Nocedal & Wright, 1999) which only requires access to an oracle solving eq. (4), i.e., sorting $\boldsymbol{\theta}$.

Alternatively, because the only term in the regularization that matters for optimization is $\boldsymbol{\alpha}$, we can regularize it alone, and directly restrict the number of non-zero entries of $\boldsymbol{\alpha}$ to some $k > 2$ as follows

$$\boldsymbol{\alpha}^{\text{top-}k\text{ sparsemax}}(\boldsymbol{\theta}) \in \underset{\boldsymbol{\alpha}\in\triangle^{|\Sigma_d|}, \|\boldsymbol{\alpha}\|_0 \leq k}{\arg\max} \quad \boldsymbol{\theta}^{\top}\mathbb{E}_{\sigma\sim\boldsymbol{\alpha}}[\boldsymbol{\rho}^{\sigma}] - \frac{\tau}{2}\|\boldsymbol{\alpha}\|_2^2, \tag{8}$$

where we assume ties are resolved arbitrarily.

This is a formulation of top-$k$ sparsemax introduced in Section 3.3 that we can efficiently employ to learn DAGs provided that we have access to a fast algorithm that returns a set of $k$ permutations with highest value of $g_{\boldsymbol{\theta}}(\sigma) = \boldsymbol{\theta}^{\top}\boldsymbol{\rho}^{\sigma}$. Algorithm 1 describes such an oracle, which, to our knowledge, has never been derived before and may be of independent interest. The algorithm restricts the search for the best solutions to the set of permutations that are one adjacent transposition away from the best solutions found so far. We refer the reader to Appendix A for notation and proof of correctness of Algorithm 1.

---

**Algorithm 1:** Top-$k$ permutations.

---
**Data:** $k \in [d!]$, $\boldsymbol{\theta} \in \mathbb{R}^d$
**Result:** top-$k$ permutations $T_k(\boldsymbol{\theta})$
$P(\boldsymbol{\theta}) \leftarrow \{\sigma^1 \in_R$
 $\arg\max_{\sigma\in\Sigma_d} g_{\boldsymbol{\theta}}(\sigma)\}$;
**while** $|T_k(\boldsymbol{\theta})| \leq k$ **do**
 $\quad \sigma \in_R$
 $\qquad \arg\max_{\sigma\in P(\boldsymbol{\theta})\setminus T_k(\boldsymbol{\theta})} g_{\boldsymbol{\theta}}(\sigma)$;
 $\quad P(\boldsymbol{\theta}) \leftarrow P(\boldsymbol{\theta}) \cup \{\sigma j \mid j \in$
 $\qquad [d-1]\}$;
 $\quad T_k(\boldsymbol{\theta}) \leftarrow T_k(\boldsymbol{\theta}) \cup \{\sigma\}$;
**end**

---

**Remark:** Top-$k$ sparsemax optimizes over the highest-scoring permutations, while sparseMAP draws any set of permutations the marginal solution can be decomposed into. As an example, for $\boldsymbol{\theta} = \mathbf{0}$, sparseMAP returns two permutations, $\sigma$ and its inverse $\sigma^{-1}$. On the other hand, because at $\mathbf{0}$ all the permutations have the same probability, top-$k$ sparsemax returns an arbitrary subset of $k$ permutations which, when using the oracle presented above, will lie on the same face of the permutahedron. In Appendix D we provide an empirical comparison of these two operators when applied to the DAG learning problem.

## 4.2 DAG LEARNING

In order to select which edges to drop from $\mathbf{R}$, we regularize the set of edge functions $\{f^{\phi_j}\}_{j=1}^d$ to be sparse via $\Omega(\boldsymbol{\Phi})$ in eq. (2), i.e., $\|\boldsymbol{\Phi}\|_0$ or $\|\boldsymbol{\Phi}\|_1$. Incorporating the sparse decompositions of eq. (7) or eq. (8) into the original problem in eq. (2) yields

$$\min_{\boldsymbol{\theta},\boldsymbol{\Phi}} \mathbb{E}_{\sigma\sim\boldsymbol{\alpha}^{\star}(\boldsymbol{\theta})}\left[\sum_{j=1}^d \ell\left(\mathbf{x}_j, f^{\phi_j}\left(\mathbf{X}\circ(\mathbf{R}^{\sigma})_j\right)\right) + \lambda\Omega(\boldsymbol{\Phi})\right], \tag{9}$$

where $(\mathbf{R}^{\sigma})_j$ is the $j$th column of $\mathbf{R}^{\sigma}$ and $\alpha^{\star}$ is the top-$k$ sparsemax or the SparseMAP distribution. Notice that for both sparse operators, in the limit $\tau \to 0_+$ the distribution $\alpha^{\star}$ puts all probability mass on one permutation: $\sigma(\boldsymbol{\theta})$ (the sorting of $\boldsymbol{\theta}$), and thus eq. (9) is a generalization of eq. (2).

We can solve the above optimization problem for the optimal $\boldsymbol{\theta}, \boldsymbol{\Phi}$ jointly via gradient-based optimization. The downside of this, however, is that training may move towards poor permutations purely because $\boldsymbol{\Phi}$ is far from optimal. Specifically, at each iteration, the distribution over permutations $\alpha^{\star}(\boldsymbol{\theta})$ is updated based on functional parameters $\boldsymbol{\Phi}$ that are, on average, good for all selected permutations. In early iterations, this approach can be highly suboptimal as it cannot escape from high-error local minima. To address this, we may push the optimization over $\boldsymbol{\Phi}$ inside the objective:

$$\min_{\boldsymbol{\theta}} \mathbb{E}_{\sigma\sim\boldsymbol{\alpha}^{\star}(\boldsymbol{\theta})}\left[\sum_{j=1}^d \ell\left(\mathbf{x}_j, f^{\phi^{\star}(\sigma)_j}\left(\mathbf{X}\circ(\mathbf{R}^{\sigma})_j\right)\right)\right] \tag{10}$$

$$\text{s.t. } \boldsymbol{\Phi}^{\star}(\sigma) = \arg\min_{\boldsymbol{\Phi}}\sum_{j=1}^d \ell\left(\mathbf{x}_j, f^{\phi_j}\left(\mathbf{X}\circ(\mathbf{R}^{\sigma})_j\right)\right) + \lambda\Omega(\boldsymbol{\Phi}).$$

This is a bi-level optimization problem (Franceschi et al., 2018; Dempe & Zemkoho, 2020) where the inner problem fits one set of structural equations $\{f^{\phi_j}\}_{j=1}^d$ per $\sigma \sim \alpha^{\star}(\boldsymbol{\theta})$. Note that, as opposed to many other settings (e.g. in meta-learning), the outer objective depends on $\boldsymbol{\theta}$ only through the distribution $\alpha_*(\boldsymbol{\theta})$, and not through the inner problem over $\boldsymbol{\Phi}$. In practice, this means that the outer

optimization does not require gradients (or differentiability) of the inner solutions at all, saving computation and allowing for greater flexibility in picking a solver for fitting $\mathbf{\Phi}^\star(\sigma)$.[2] For example, for $\Omega(\mathbf{\Phi}) = \|\mathbf{\Phi}\|_1$ we may invoke any Lasso solver, and for $\Omega(\mathbf{\Phi}) = \|\mathbf{\Phi}\|_0$ we may use the algorithm of Louizos et al. (2017), detailed in Appendix B. The downside of this bi-level optimization is that it is only tractable when the support of $\alpha^\star(\boldsymbol{\theta})$ has a few permutations. Optimizing for $\boldsymbol{\theta}$ and $\mathbf{\Phi}$ jointly is more efficient.

### 4.3 COMPUTATIONAL ANALYSIS

The overall complexity of our framework depends on the choice of sparse operator for learning the topological order and on the choice of the estimator for learning the edge functions. We analyze the complexity of learning topological orderings specific to our framework, and refer the reader to previous works for the analyses of particular estimators (e.g., Efron et al. (2004)). Note that, independently from the choice of sparse operator, the space complexity of DAGuerreotype is at least of the order of the edge masking matrix $\mathbf{R}$, hence $O(d^2)$. This is in line with most methods based on continuous optimization and can be improved by imposing additional constraints on the in-degree and out-degree of a node.

**SparseMAP.** Each SparseMAP iteration involves a length-$d$ argsort and a Cholesky update of a $s$-by-$s$ matrix, where $s$ is the size of the active set (number of selected permutations), and by Carathéodory's convex hull theorem (Reay, 1965) can be bounded by $d + 1$. Given a fixed number of iterations $K$ (as in our implementation), this leads to a time complexity of $\mathcal{O}(Kd^2)$ and space complexity $\mathcal{O}(s^2 + sd)$ for SparseMAP. Furthermore, we warm-start the sorting algorithm with the last selected permutation. Both in theory and in practice, this is better than the $\mathcal{O}(d^3)$ complexity of maximization over the Birkhoff polytope.

**Top-$k$ sparsemax.** Complexity for top-$k$ sparsemax is dominated by the complexity of the top-k oracle. In our implementation, the top-k oracle has an overall time complexity $\mathcal{O}(K^2d^2)$ and space complexity $\mathcal{O}(Kd^2)$ (as detailed in Appendix A) when searching for the best $K$ permutations. When $K$ is fixed, as in our implementation, this leads to an overall complexity of the top-k sparsemax operator $\mathcal{O}(d^2)$. In practice, $K$ has to be of an order smaller than $\sqrt{d}$ for our framework to be more efficient than existing end-to-end approaches.

### 4.4 RELATIONSHIP TO PREVIOUS DIFFERENTIABLE ORDER-BASED METHODS

The advantages of our method over Cundy et al. (2021); Charpentier et al. (2022) are: (a) our parametrization, based on sorting, improves efficiency in practice (Appendix D) and has theoretically stabler learning dynamics as measured by our bound on SHD (Theorem (C.1)); (b) our method allows for any downstream edge estimator, including non-differentiable ones, critically allowing for off-the-shelf estimators; (c) empirically our method vastly improves over both approaches in terms of SID and especially SHD on both real-world and synthetic data (Section 5).

## 5 EXPERIMENTS

### 5.1 EXPERIMENTAL SETUP

**Datasets** Reisach et al. (2021) recently demonstrated that commonly studied synthetic benchmarks have a key flaw. Specifically, for linear additive synthetic DAGs, the marginal variance of each node increases the 'deeper' the node is in the DAG (i.e., all child nodes generally have marginal variance larger than their parents). They empirically show that a simple baseline that sorts nodes by increasing marginal variance and then applies sparse linear regression matches or outperforms state-of-the-art DAG learning methods. Given the triviality of simulated DAGs, here we evaluate all methods on two real-world tasks: *Sachs* (Sachs et al., 2005), a dataset of cytometric measurements of phosphorylated protein and phospholipid components in human immune system cells. The problem consists of $d = 11$

---

[2]This computational advantage is shared with the score-function estimator (SFE, Rubinstein, 1986; Williams, 1992; Paisley et al., 2012; Mohamed et al., 2020). We are not aware of any applications of SFE to permutation learning, likely due to the #P-completeness of marginal inference over the Birkhoff polytope (Valiant, 1979; Taskar, 2004).

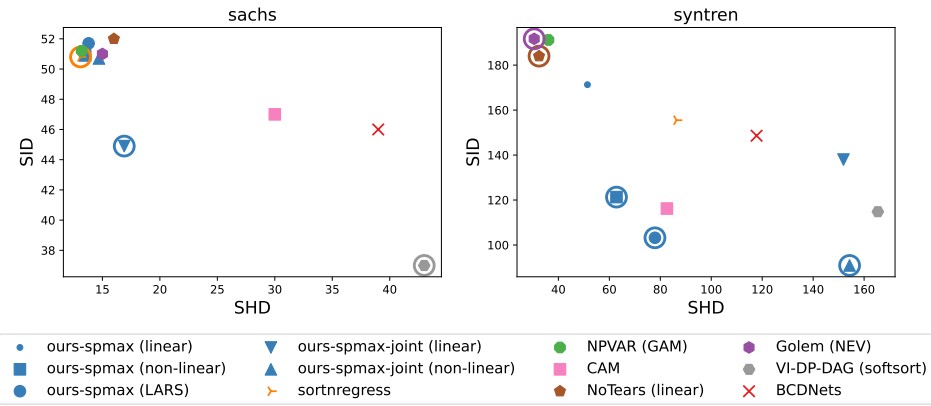

Figure 2: SHD vs SID on real datasets *Sachs* and *SynTReN*. Results are averaged over 10 seeds and the solutions lying on the Pareto front are circled.

nodes, 853 observations and of the graph reconstructed by Sachs et al. (2005) as ground-truth DAG, which contains 17 edges; *SynTReN* (den Bulcke et al., 2006), a set of 10 pseudo-real transcriptional networks generated by the SynTRen simulator, each consisting of 500 simulated gene expression observations, and a DAG of $d = 20$ nodes and of $e$ edges with $e \in \{20, \dots, 25\}$. We use the networks made publicly available by Lachapelle et al. (2020). In Appendix D we, however, compare different configurations of our method also on synthetic datasets, as they constitute an ideal test-bed for assessing the quality of the ordering learning step independently from the choice of structural equation estimator.

**Baselines** We benchmark our framework against state-of-the-art methods: *NoTears* (both its linear (Zheng et al., 2018) and nonlinear (Zheng et al., 2020) models), the first continuous optimization method, which optimizes the Frobenius reconstruction loss and where the DAG constraint is enforced via the Augmented Lagrangian approach; *Golem* (Ng et al., 2020), another continuous optimization method that optimizes the likelihood under Gaussian non-equal variance error assumptions regularized by *NoTears*'s DAG penalty; *CAM* (Bühlmann et al., 2014), a two-stage approach that estimates the variable order by maximum likelihood estimation based on an additive structural equation model with Gaussian noise; *NPVAR* (Gao et al., 2020), an iterative algorithm that learns topological generations and then prunes edges based on node residual variances (with the Generalized Additive Models (Hastie & Tibshirani, 2017) regressor backend to estimate conditional variance); *sortnregress* (Reisach et al., 2021), a two-steps strategy that orders nodes by increasing variance and selects the parents of a node among all its predecessors using the Least Angle Regressor (Efron et al., 2004); *BCDNets* (Cundy et al., 2021) and *VI-DP-DAG* (Charpentier et al., 2022), the two differentiable, probabilistic methods described in Section 2. Before evaluation, we post-process the graphs found by *NoTears* and *Golem* by first removing all edges with absolute weights smaller than 0.3 and then iteratively removing edges ordered by increasing weight until obtaining a DAG, as the learned graphs often contain cycles.

**Metrics** We compare the methods by two metrics, assessing the quality of the estimated graphs: the Structural Hamming Distance (*SHD*) between true and estimated graphs, which counts the number of edges that need to be added or removed or reversed to obtain the true DAG from the predicted one; and the Structural Intervention Distance (*SID*, Peters & Bühlmann, 2015), which counts the number of causal paths that are broken in the predicted DAG. It is standard in the literature to compute both metrics, given their complementarity: SHD evaluates the correctness of individual edges, while SID evaluates the preservation of causal orderings. We further remark here that these metrics privilege opposite trivial solutions. Because true DAGs are usually sparse (i.e., their number of edges is much smaller than the number of possible ones), SHD favors sparse solutions such as the empty graph. On the contrary, given a topological ordering, SID favors dense solutions (complete DAGs in the limit) as they are less likely to break causal paths. For this reason, we report both metrics and highlight the solutions on the Pareto front in Figure 2.

**Hyper-parameters and training details** We set the hyper-parameters of all methods to their default values. For our method we tuned them by Bayesian Optimization based on the performance, in terms

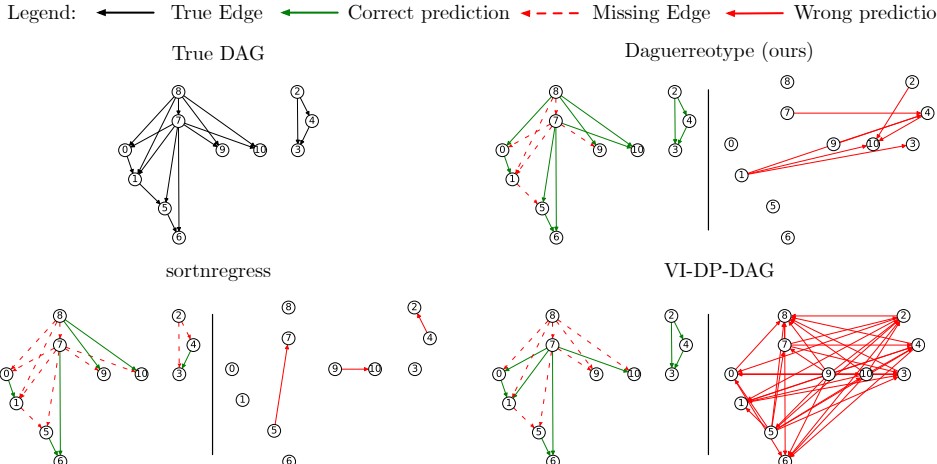

Figure 3: *Sachs*. True DAG and DAGs predicted by the best-performing methods. We plot on the left of the bar correct and missing edges and on the right of the bar wrong edges found by each method. DAGuerreotype strikes a good balance between SID and SHD; other methods focus overly on one over the other by either predicting too few (sortnregress) or too many edges (VI-DP-DAG).

of SHD and SID, averaged over several synthetic problems from different SEMs. For our method, we optimize the data likelihood (under Gaussian equal variance error assumptions, as derived in Ng et al. (eq. 2, 2020)) and we instantiate $f_j^{\phi_j}$ to a masked linear function (linear) or as a masked MLP as for *NoTears-nonlinear*. Because our approach allows for modular solving of the functional parameters $\Phi$, we also experiment with using Least Angle Regression (LARS) (Efron et al., 2004). We additionally apply $l_2$ regularizations on $\{\theta, \Phi\}$ to stabilize training, and we standardize all datasets to ensure that all variables have comparable scales. More details are provided in Appendix D. The code for running the experiments is available at https://github.com/vzantedeschi/DAGuerreotype.

## 5.2 Results

We report the main results in Figure 2, where we omit some baselines (e.g., DAGuerreotype with sparseMAP) for sake of clarity. We present the complete comparison in Appendix D, together with additional metrics and studies on the sensitivity of DAGuerreotype depending on the choice of key hyper-parameters. To provide a better idea of the learned graphs, we also plot in Figure 3 the graphs learned by the best-performing methods on *Sachs*. We observe that the solutions found by NPVAR and the matrix-exponential regularized methods (NoTears and Golem) are the best in terms of SHD, but have the worst SIDs. This can be explained by the high sparsity of their predicted graphs (see the number of edges in Tables 1 and 2). On the contrary, the high density of the solutions of VI-DP-DAG makes them among the best in terms of SID and the worst in terms of SHD. DAGuerreotype provides solutions with a good trade-off between these two metrics and which lie on the Pareto front. Inevitably its performance strongly depends on the choice of edge estimator for the problem at hand. For instance, the linear estimator is better suited for *Sachs* than for *SynTReN*. In Appendix D, we assess our method's performance independently from the quality of the estimator with experiments on synthetic data where the underlying SEM is known, and the estimator can be chosen accordingly.

## 6 Conclusion, Limitations and Future Work

In this work, we presented *DAGuerreotype*, a permutation-based method for end-to-end learning of directed acyclic graphs. While our approach shows promising results in identifying DAGs, the optimization procedure can still be improved. Alternative choices of estimators, such as Generalized Additive Models (Hastie & Tibshirani, 2017) as done in CAM and NPVAR, could be considered to improve identification. Another venue for improvement would be to include interventional datasets at training. It would be interesting to study in this context whether our framework is more sample-efficient, i.e., allows us to learn the DAG with fewer interventions or observations.

## ACKNOWLEDGEMENTS

We are grateful to Mathieu Blondel, Caio Corro, Alexandre Drouin and Sébastien Paquet for discussions. Part of this work was carried out when VZ was affiliated with INRIA-London and University College London. Experiments presented in this paper were partly performed using the Grid'5000 testbed, supported by a scientific interest group hosted by Inria and including CNRS, RENATER and several Universities as well as other organizations (see https://www.grid5000.fr). VN acknowledges support from the Dutch Research Council (NWO) project VI.Veni.212.228.

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

## A  TOP-K ORACLE

In this section, we propose an efficient algorithm for finding the top-k scoring rankings of a vector and prove its correctness. We leverage this result in order to apply the sparsemax operator (Correia et al., 2020) to our DAG learning problem. We are not aware of existing works on this topic and believe this result is of independent interest.

### A.1  NOTATION AND DEFINITIONS

Let us denote the objective function $g_{\boldsymbol{\theta}}(\sigma) = \langle \boldsymbol{\theta}, \boldsymbol{\rho}^\sigma \rangle$ evaluating the quality of a permutation $\sigma$, and denote one of its maximizers as $\sigma^1 \in \arg\max_{\sigma \in \Sigma_d} g_{\boldsymbol{\theta}}(\sigma)$, corresponding to the argsort of $\boldsymbol{\theta}$. Notice that we can equivalently write this objective as $g_{\boldsymbol{\theta}}(\sigma) = \langle \boldsymbol{\theta}^\sigma, \boldsymbol{\rho} \rangle$ by applying the permutation to $\boldsymbol{\theta}$.

**Definition A.1** (Top-$k$ permutations). *We denote by $T_k(\boldsymbol{\theta}) \subseteq \Sigma_d$ a sequence of K highest-scoring permutations according to $g_{\boldsymbol{\theta}}$ i.e., $T_k(\boldsymbol{\theta}) = \{\sigma^1, \sigma^2, \ldots, \sigma^k\}$ and for any $\sigma' \notin T_k(\boldsymbol{\theta})$:*

$$g_{\boldsymbol{\theta}}(\sigma^1) \geq g_{\boldsymbol{\theta}}(\sigma^2) \geq \ldots \geq g_{\boldsymbol{\theta}}(\sigma^k) \geq g_{\boldsymbol{\theta}}(\sigma').$$

In this section, we will make use of the following definition and lemma.

**Definition A.2** (Adjacent transposition). *$\sigma j := \sigma \, (j \; j{+}1)$ denotes a 2-cycle of the components of the vector $\sigma$, which transposes (flips) the two consecutive elements $\sigma_j$ and $\sigma_{j+1}$.*

**Lemma A.1.** *Let $\sigma \in \Sigma_d$ be a permutation. Exactly one of the following holds.*

1. *$g_{\boldsymbol{\theta}}(\sigma \, j) = g_{\boldsymbol{\theta}}(\sigma^1)$,*

2. *There exists an adjacent transposition $(j \; j{+}1)$ such that $\sigma j$ satisfies $g_{\boldsymbol{\theta}}(\sigma) > g_{\boldsymbol{\theta}}(\sigma^1)$.*

*Proof.* Denote by $\boldsymbol{\theta}^\sigma$ the permutation of $\boldsymbol{\theta}$ by $\sigma$ and let $J = \{j \in [d{-}1] \mid \boldsymbol{\theta}_j^\sigma > \boldsymbol{\theta}_{j+1}^\sigma\}$. If $J = \emptyset$ then $\boldsymbol{\theta}^\sigma$ is in increasing order, so by the Rearrangement Inequality (Hardy et al., 1952, Thms. 368–369) $\sigma$ is a 1-best permutation. Otherwise, applying the adjacent transposition $(j \; j{+}1)$ increases the score:

$$g_{\boldsymbol{\theta}}(\sigma \, j) - g_{\boldsymbol{\theta}}(\sigma) = \boldsymbol{\theta}_j^\sigma - \boldsymbol{\theta}_{j+1}^\sigma > 0 \, .$$

$\square$

### A.2  BEST-FIRST SEARCH ALGORITHM

Algorithm (2) finds the set of $k$-best scoring permutations given $\boldsymbol{\theta}$. Starting from an optimum of $g_{\boldsymbol{\theta}}$, the algorithm grows the set of candidate permutations $P(\boldsymbol{\theta})$ by adding all those that are one adjacent transposition away from the $i$th-best permutation at iteration $i$. It then selects the best scoring permutation in $P(\boldsymbol{\theta})$ to be the top-$(i{+}1)$ solution. Theorem A.2 proves that an $i{+}1$th-best solution must lie in this set $P(\boldsymbol{\theta})$, hence that Algorithm (2) is correct.

**Theorem A.2** (Correctness of Algorithm (2)). *Given a sequence of $k{-}1$-best permutations $T_{k-1}(\boldsymbol{\theta}) = \{\sigma^1, \sigma^2, \ldots, \sigma^{k-1}\}$, there exists a $k$-th best $\sigma^k$, i.e., one satisfying $g_{\boldsymbol{\theta}}(\sigma^{k-1}) \geq g_{\boldsymbol{\theta}}(\sigma^k) \geq g_{\boldsymbol{\theta}}(\sigma')$ for any $\sigma' \notin T_{k-1}(\boldsymbol{\theta})$, with the property that $\sigma^k = \sigma^i j$ for some $i \in \{1, \ldots, k{-}1\}$ and $j \in \{1, \ldots, d{-}1\}$.*

*Proof.* Let $\sigma^k$ be a k-th best permutation.

Case 1. $g_{\boldsymbol{\theta}}(\sigma^k) \neq g_{\boldsymbol{\theta}}(\sigma^1)$. Invoking Lemma A.1 we have $g_{\boldsymbol{\theta}}(\sigma^k j) > g_{\boldsymbol{\theta}}(\sigma^k)$. Since the inequality is strict, we must have $\sigma^k j \in T_{k-1}(\boldsymbol{\theta})$.

Case 2. $g_{\boldsymbol{\theta}}(\sigma^k) = g_{\boldsymbol{\theta}}(\sigma^1)$. In this case, we have at least $k$ permutations tied for first place. Any two permutations with equal score can only differ in indices that correspond to ties in $\boldsymbol{\theta}$. Therefore, any two tied permutations are connected by a trajectory of transpositions of equal score. This is a face of the permutahedron, of size $c > K$, containing $T_{k-1}(\boldsymbol{\theta})$. This face must contain at least one permutation that is one adjacent transposition away from one of $T_{k-1}(\boldsymbol{\theta})$, and we may as well take this one as the $k$-th best instead of $\sigma^k$. $\square$

**Algorithm 2:** Top-$k$ permutations.

---

**Data:** $k \in \{1, \ldots, d!\}, \boldsymbol{\theta} \in \mathbb{R}^d$
**Result:** top-$k$ permutations $T_k(\boldsymbol{\theta})$
$P(\boldsymbol{\theta}) \leftarrow \{\sigma^1 \in_R \arg\max_{\sigma \in \Sigma_d} g_{\boldsymbol{\theta}}(\sigma)\}$ /* initialize set of candidates
   with an optimum                                                                */
**while** $|T_k(\boldsymbol{\theta})| \leq k$ **do**
   $\sigma \in_R \arg\max_{\sigma \in P(\boldsymbol{\theta}) \setminus T_k(\boldsymbol{\theta})} g_{\boldsymbol{\theta}}(\sigma)$ /* retrieve a best scoring solution
      among the candidates that has not been retrieved yet    */
   $P(\boldsymbol{\theta}) \leftarrow P(\boldsymbol{\theta}) \cup \{\sigma j \mid j \in \{1, \ldots, d-1\}\}$ /* add its one adjacent
      transposition away permutations to the candidates     */
   $T_k(\boldsymbol{\theta}) \leftarrow T_k(\boldsymbol{\theta}) \cup \{\sigma\}$ /* update set of best permutations    */
**end**

---

### A.3 COMPUTATIONAL ANALYSIS

Finding $\sigma^1$ requires sorting a vector of size $d$ (hence $O(d \log d)$ complexity). Then, at each iteration $k$ of Algorithm (2), the maximum among the best candidates $P(\boldsymbol{\theta})$ needs to be found, which requires $O(dk)$ complexity as $|P(\boldsymbol{\theta})| \leq (d-2)k$, and $O(d)$ adjacent flip operations are applied to it.

The most expensive operation is checking that the selected best candidate is not already in $T_{k-1}(\boldsymbol{\theta})$. At worst, this requires going through the whole $P(\boldsymbol{\theta})$ and comparing them in order to all top-k solutions, so no more than $K \times |P(\boldsymbol{\theta})|$ comparisons of cost $O(d)$ each.

This leads to an overall time complexity of $O(K^2 d^2)$. The space complexity is dominated by the size of $P(\boldsymbol{\theta})$, which contains at most $Kd$ vectors of size $d$, leading to $O(Kd^2)$.

## B  L0 REGULARIZATION

In the linear and non-linear variants of DAGuerreotype , we implement the regularization term $\Omega_\xi$ of the inner problem of Eq. (10) with an approximate L0 regularizer. The exact L0 norm

$$||\mathbf{w}||_0 = \sum_{i=1}^n \mathbf{1}_{w_j \neq 0} \tag{11}$$

counts the number of non-zero entries of $\mathbf{w} \in \mathbb{R}^n$ and, when used as regularizer, it favors sparse solutions without injecting other priors. However, its combinatorial nature and its non-differentiability make its optimization intractable. Following Louizos et al. (2017), we reparameterize (11) by introducing a set of binary variables $\mathbf{z} \in \{0,1\}^n$ and letting $\mathbf{w} = \hat{\mathbf{w}} \circ \mathbf{z}$, so that $||\mathbf{w}||_0 = \sum z_i$. Next, we let $\mathbf{z} \sim p(\mathbf{z}; \boldsymbol{\pi}) = \text{Bernoulli}(\boldsymbol{\pi})$ where $\boldsymbol{\pi} \in [0,1]^d$. For linear SEMs, we can now reformulate the Inner Problem (10) as follows:

$$\min_{\hat{\mathbf{w}}, \boldsymbol{\pi}} \sum_{j=1}^d \left( \mathbb{E}_{z_j \sim p(z_j | \pi_j)} \left[ \ell \left( \mathbf{x}_j, f_j^{\hat{w}_j \circ z_j} \left( \mathbf{X}, (\mathbf{M}^\sigma)_j \right) \right) \right] + \xi \sum_{i=1}^d \pi_{ji} \right); \tag{12}$$

where now the decision (inner) variables are intended to be matrices and $\xi \geq 0$ is a hyperparameter. In the non-linear (MLP) case, we achieve sparsity at the graph level by group-regularizing the parameters corresponding to each input variable. In the experiments, we optimize (12) using the one-sample Monte Carlo straight-through estimator and set the final functional parameters as

$$\mathbf{w}_* = \hat{\mathbf{w}}_* \circ \text{MAP}\left[p(\,\cdot\,; \boldsymbol{\pi})\right] = \hat{\mathbf{w}}_* \circ H \left( \boldsymbol{\pi} - \frac{1}{2}\mathbf{1} \right); \tag{13}$$

where $H$ is the Heaviside function. We leave the implementation of more sophisticated strategies, such as the relaxation with the hard-concrete distribution presented in (Louizos et al., 2017) or other estimators (e.g. Paulus et al., 2020; Niepert et al., 2021), to future work.

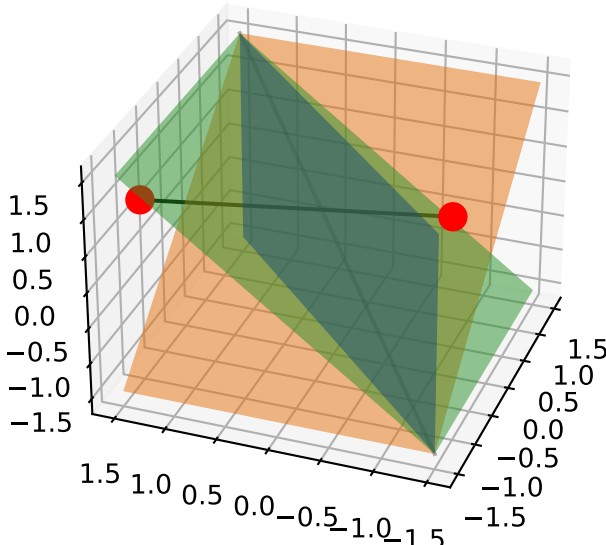

Figure 4: Representation of the degeneracy hyper-planes in $\mathbb{R}^3$ and of two-parameter points (in red) whose connecting segment intersects two hyper-planes.

## C  CHARACTERIZATION AND SENSITIVITY OF THE VECTOR PARAMETRIZATION

In this section, we provide some intuition into the behavior of our score vector parameterization in the space of complete DAGs. More precisely, we look at the Maximum A Posteriori (MAP) complete DAG

$$M^{\sigma^*} \in \mathbb{D}_C[d] \quad \text{where} \quad \sigma^* \in \arg\max \langle \boldsymbol{\theta}, \boldsymbol{\rho}_\sigma \rangle \tag{14}$$

and how it varies as a function of the score vector $\boldsymbol{\theta} \in \mathbb{R}^d$. The permutation $\sigma^*$ is a MAP state (or mode) of both the sparseMAP and the sparsemax distributions (as well as the standard categorical/softmax distribution). For brevity, we shall rename $M^{\sigma^*} := M(\boldsymbol{\theta})$ in the following.

We choose to work on the space of complete DAGs because there is a one-to-one correspondence between topological orderings and complete DAGs. This is not generally true when analyzing the space of DAGs, as a permutation does not uniquely identify a DAG and vice versa.

### C.1  PRELIMINARY

For the results of this section, we will use the following relationship between transpositions (adjacent or not) and SHD.

**Proposition C.0.1** (SHD difference after a flip). *Consider $\boldsymbol{\theta} \in \mathbb{R}^d$ and $\boldsymbol{\theta}'$ which is obtained by applying a flip $(i\ j)$ with the convention that $i < j$. All the edges from nodes between $i$ and $j$ directed towards $i$ or $j$ need to be reversed. Thus, the SHD between complete DAGs $\mathrm{SHD}(M(\boldsymbol{\theta}), M(\boldsymbol{\theta}')) = 2(j-i) - 1$. (because no new undirected edges are added, no undirected edges are removed, and $2(j-i) - 1$ edges are reversed).*

From Proposition (C.0.1) we deduce that the SHD difference after applying an adjacent flip is $\mathrm{SHD}(M(\boldsymbol{\theta}), M(\boldsymbol{\theta}')) = 1$.

### C.2  ANALYSIS

Recall that $\sigma^*$ sorts the elements of $\boldsymbol{\theta}$ in increasing order. Then, the points $\boldsymbol{\theta} \in \mathbb{R}^d$ where $\arg\max \langle \boldsymbol{\theta}, \boldsymbol{\rho}_\sigma \rangle$ is non-singleton (degeneracy) are exactly those that have at least one tie among their

entries (i.e., $\exists\, i,j$ such that $\boldsymbol{\theta}_i = \boldsymbol{\theta}_j$). Following this simple observation, we can populate $\mathbb{R}^d$ with $\binom{d}{2}$ hyper-planes (of dimensionality $d-1$), and call them $H_{i,j}$ with $i < j$, such that $\forall \boldsymbol{\theta} \in H_{i,j}, \boldsymbol{\theta}_i = \boldsymbol{\theta}_j$. Intersections of such hyper-planes are lower dimensional subspaces where more than 2 entries of $\boldsymbol{\theta}$ are equal. For $d = 3$, Figure 4 depicts the $\binom{3}{2} = 3$ hyper-planes in orange, green and blue, and their intersection (a line) in gray.

The hyper-planes $\{H_{i,j}\}_{i<j}$'s delimit exactly $d!$ open cones of the form $C = \{\boldsymbol{\theta} \in \mathbb{R}^d | \boldsymbol{\theta}_1^\sigma < \boldsymbol{\theta}_2^\sigma < \cdots < \boldsymbol{\theta}_d^\sigma, \sigma \in \Sigma_d\}$, i.e., each cone is the space of all points that give the same MAP permutation and do not contain any ties.

This partition of the space allows us to reason about the sensitivity of our MAP estimates to changes to the parameters. For instance, as the points of a cone do not have any ties, changes to the score vector do not affect SHD: $\forall \boldsymbol{\theta} \in C, \boldsymbol{\theta}' \in C \; \mathrm{SHD}(M(\boldsymbol{\theta}), M(\boldsymbol{\theta}')) = 0$.

We first analyze the sensitivity to single-entry changes, hence assessing how much an entry of $\boldsymbol{\theta}$ can be individually perturbed without changing its MAP. Proposition C.0.2 provides the entry-wise ranges in which the optimal solution set does not change.

**Proposition C.0.2** (Entry-wise intervals). *For any* $\boldsymbol{\theta} \in \mathbb{R}^d$ *and* $\varepsilon_i \in \mathbb{R}$, $\arg\max \langle \boldsymbol{\theta}^\sigma, \boldsymbol{\rho} \rangle = \arg\max \langle \boldsymbol{\theta}^\sigma + \varepsilon_i \mathbf{e}_i, \boldsymbol{\rho} \rangle$ *if and only if:*

- $\forall i \in [2, \ldots, d-1]$, $\varepsilon_i \in (\boldsymbol{\theta}_{i-1}^{\sigma^\star} - \boldsymbol{\theta}_i^{\sigma^\star}, \boldsymbol{\theta}_{i+1}^{\sigma^\star} - \boldsymbol{\theta}_i^{\sigma^\star})$;

- $i = 1$, $\varepsilon_i \in (-\infty, \boldsymbol{\theta}_{i+1}^{\sigma^\star} - \boldsymbol{\theta}_i^{\sigma^\star})$;

- $i = d$, $\varepsilon_i \in (\boldsymbol{\theta}_{i-1}^{\sigma^\star} - \boldsymbol{\theta}_i^{\sigma^\star}, \infty)$

*where* $\mathbf{e}_i$ *denotes the $i$-th standard unit vector and* $\sigma^\star \in \arg\max \langle \boldsymbol{\theta}, \boldsymbol{\rho}^\sigma \rangle$.

We can relate this result to changes in terms of SHD, by making the following observation: if we gradually increase one coordinate $\boldsymbol{\theta}_i$ initially ranked $\sigma_i^\star$, its rank changes in the optimal ordering as soon as it becomes greater than the coordinate right after it in the ordering, entailing an adjacent transposition between the two coordinates[3]. Greater perturbations entail a longer sequence of adjacent transpositions. By Proposition C.0.1, this implies an increase in SHD of 1 for each transposition. Visually, the SHD increases by 1 every time the perturbed vector crosses a hyper-plane along its perturbation direction, as this corresponds to swapping two components that are adjacent when optimally sorted.

For comparison, we can apply the same reasoning to the matrix parametrization of the linear assignment problem, deployed by Cundy et al. (2021) for learning permutation matrices. In this context, we can leverage the edge sensitivity analysis reviewed in Michael et al. (2020, Equations 6-7). As a general remark, it is not as intuitive to determine the entry-wise intervals for this problem, not only because we have $d^2$ variables instead of $d$ but especially because it requires solving a different assignment problem per entry. Furthermore, the minimal perturbation that changes the MAP does not necessarily result in flipping two adjacent nodes, entailing an SHD relative to the optimal permutation of at least 1. Consider, for example, the following $3 \times 3$ matrix parameter:

$$\boldsymbol{\Theta} = \begin{pmatrix} 16 & 16 & 15 \\ 5 & 16 & 10 \\ 16 & 9 & 10 \end{pmatrix}$$

Its optimal permutation matrix corresponds to the rank $(3, 1, 2)$ with a score of 29. The minimal perturbations on $\boldsymbol{\Theta}_{1,3}$ or $\boldsymbol{\Theta}_{3,2}$ that change the MAP result in the solution $(2, 1, 3)$ which has an SHD of 3 w.r.t. the optimal (and the second best score of 31). For problems of scale larger than this example, the resulting SHD can take values up to $2d - 3$ for non-adjacent transpositions.

We now generalize and formalize this relationship between perturbations and SHD changes for general vector perturbations (global sensitivity). Theorem (C.1) upper bounds the SHD between complete DAGs of any pair of score vectors by the number of hyper-planes crossed by the segment connecting them.

---

[3]A similar reasoning also applies when decreasing the value of one coordinate instead.

**Theorem C.1** (Global sensitivity). *For any $\boldsymbol{\theta} \in \mathbb{R}^d$ and $\boldsymbol{\theta}' \in \mathbb{R}^d$*

$$\mathrm{SHD}(M(\boldsymbol{\theta}), M(\boldsymbol{\theta}')) \leq \int_{t \in [0,1]} \sum_i \sum_{j > i} \delta_{H_{i,j}}(\boldsymbol{\theta} + t(\boldsymbol{\theta}' - \boldsymbol{\theta})) \, \mathrm{d}t \tag{15}$$

*where $\delta_A(x)$ is the (generalized) Dirac delta that evaluates to infinity if $x \in A$ and 0 otherwise.*

*Proof.* Let us first consider the case where $\boldsymbol{\theta} \in C$ and $\boldsymbol{\theta}' \in C'$, i.e., the two vectors do not contain ties. Let us denote $\sigma$ (respectively $\sigma'$) the permutation that sorts the components of a point of $C$ ($C'$) by increasing order. The minimal-length sequence of adjacent flips that need to be applied to $\sigma$ to obtain $\sigma'$ has a length equal to the number of times the segment connecting their parameters crosses a hyper-plane. Then the SHD between their complete DAGs equals the minimal number of adjacent flips, which proves the result. When either $\boldsymbol{\theta}$ or $\boldsymbol{\theta}'$ lies on a hyper-plane, the minimal required number of adjacent flips might be smaller, hence the upper bound in Theorem C.1. $\qquad\square$

In Figure 4, the segment connecting the two red dots intersects the green and blue hyperplanes and hence the resulting complete DAGs will have an SHD of at most 2 (in fact, exactly 2 in this case).

This intuitive characterization links the SHD distance in the complete DAG space to a partition of $\mathbb{R}^d$ resulting from the "sorting" operator (i.e. the MAP, or maximizer of the linear program) of the score vector $\boldsymbol{\theta}$. This implies that, during optimization, if $\boldsymbol{\theta}_k$ are in the interior of any cone (which happens almost surely) then it is very likely that updating the parameters results in small changes to the SHD unless the parameters have all similar values.

A deeper analysis of the vector parameterization is the object of future work, as we hope it can fuel further improvements in the optimization algorithm, such as better initialization strategies or reparameterization. For instance, optimizing in $\mathbb{S}_r^{d-1}$ (over polar coordinates) would expose the role of the radius $r$ as similar to the temperature parameter for the distributions we consider (the smaller the radius, the higher the "temperature"). Typically, the temperature is left constant during training or annealed, suggesting this might be advantageous also in our scenario. We leave the exploration of this strategy to future work.

## D  ADDITIONAL EXPERIMENTS

We provide a detailed description of the experimental setup and report additional results. The method is implemented in (PyTorch, Paszke et al., 2019), and the code used for carrying out the experiments is included in the supplementary material. All experiments were run on a machine with 16 cores, $32Gb$ of RAM and an NVIDIA A100-SXM4-80GB GPU.

**DAGuerreotype 's optimization and evaluation** For our method, we optimize the data likelihood (under Gaussian equal variance error assumptions, as derived in Ng et al. (Equation 2, 2020)) We optimize the bilevel Problem (10) when not specified otherwise. We report results for the following three variants of DAGuerreotype , where the edge estimator of the graph is instantiated

- (linear)   with L0 regularization, $f_j^{\boldsymbol{\phi}_j}(\mathbf{X}, \mathbf{A}_j) = \mathbf{X}(\boldsymbol{\phi}_j \circ \mathbf{A}_j)$ and $w_j \in \mathbb{R}^d$;

- (non-linear)   with L0 regularization, $f_j^{\boldsymbol{\phi}_j}(\mathbf{X}, \mathbf{A}_j) = h_j\left(g_j^{\boldsymbol{\phi}_j}(\mathbf{X}, \mathbf{A}_j)\right)$, with $h_j$ a locally connected MLP with one hidden layer, 50 hidden units and sigmoid activation function, $g_j^{\boldsymbol{\phi}_j}$ a linear layer with $d \times 50$ hidden units (50 per parent $i$) and masking out all non-parents of $j$ (according to $\mathbf{A}_j$), and $\boldsymbol{\phi}_{ji} = \|g_{ji}^{\boldsymbol{\phi}_j}\|_2$ as for *NoTears* (non-linear);

- (LARS)   with Least Angle Regression (LARS) (Efron et al., 2004), $f_j^{\boldsymbol{\phi}_j}(\mathbf{X}, \mathbf{A}_j) = \mathbf{X}(\boldsymbol{\phi}_j \circ \mathbf{A}_j)$ and $\boldsymbol{\phi}_j \in \mathbb{R}^d$.

We additionally apply a $l_2$ regularization on $\{\boldsymbol{\theta}, \boldsymbol{\phi}\}$ to stabilize training, and we standardize all datasets to ensure that all variables have comparable scales.

With any variant, the outer problem is optimized for $5,000$ maximum iterations by gradient descent and early-stopped when approximate convergence is reached. When using the (linear) and (non-linear)

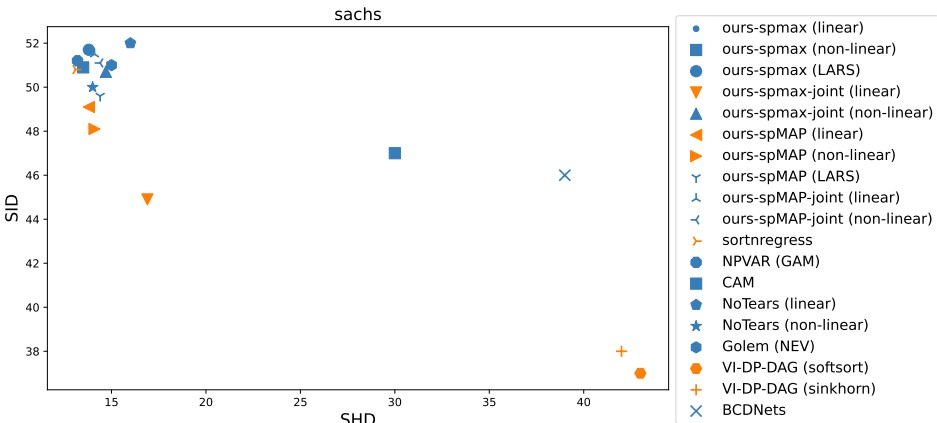

Figure 5: SHD vs SID on *Sachs*. The solutions lying on the Pareto front are colored in orange and the others in blue.

back-ends, the graph of each permutation is optimized also by gradient descent, for $1,000$ epochs and also with an early-stopping mechanism based on approximate convergence. After training, the graph for the mode permutation is further fine-tuned, and the final evaluation is carried out with this model.

We also experiment with the approximated, but faster, joint optimization of $\theta$ and $\phi$ in Problem (2) instead of the bilevel formulation, when we add the suffix *joint* to the method name. In this case, as we need a differentiable graph estimator for jointly updating the ordering and the graph, we instantiate the method only with the (linear) and (non-linear) back-ends. These models are optimized by gradient descent for $5,000$ epochs and with an early-stopping mechanism based on approximate convergence.

The default hyper-parameters of our methods were chosen as follows. We set the sparse operators' temperature $\tau = 1$ and $K = 100$, the strength of the $l_2$ regularizations to $0.0005$, and tuned the learning rates for the outer and inner optimization $\in [10^{-4}, 10^{-1}]$ and pruning strength $\lambda \in [10^{-6}, 10^{-1}]$. The tuning was carried out by Bayesian Optimization using (Optuna, Akiba et al., 2019) for $50$ trials on synthetic problems, consisting of data generated from different types of random graphs (Scale-Free, Erdős–Rényi, BiPartite) and of noise models (e.g., Gaussian, Gumbel, Uniform) with $20$ nodes and $20$ or $40$ expected edges. For each setting, three datasets are generated by drawing a DAG, its edge weights uniformly in $[-2, -0.5] \cup [0.5, 2]$, and $1,000$ data points. A set of hyper-parameters is then evaluated by averaging its performance on all the generated datasets, and the tuning is carried out to minimize SHD and SID jointly. The default value of a hyper-parameter was then set to be the average value among those lying on the Pareto front and rounded up to have a single significant digit.

**Baseline optimization and evaluation**

All baseline methods are optimized using the codes released by their authors, apart from *CAM* that is included in the Kalainathan & Goudet (Causal Discovery Toolbox, 2019). Before evaluation, we post-process the graphs found by *NoTears* and *Golem* by first removing all edges with absolute weights smaller than $0.3$ and then iteratively removing edges ordered by increasing weight until obtaining a DAG, as the learned graphs often contain cycles. For the probabilistic baselines (*VI-DP-DAG* and *BCDNets*), we make use of the mode model (in particular, the mode permutation matrix) for evaluation.

We set the hyper-parameters of all methods to their default values, released together with the source code, apart from the parameters of the Least Angle Regressor module of *sortnregress* that uses the Bayesian Information Criterion for model selection.

**Additional results on real-world tasks** In Figures 5, 6, and Tables 1 and 2 we extend the evaluation on real-world tasks provided in the main text. More precisely, we report the results also for DAGuerreotype with sparseMAP, and provide additional metrics for comparison: the *F1* score using the existence of an edge as the positive class and the number of predicted edges to assess the density of the solutions.

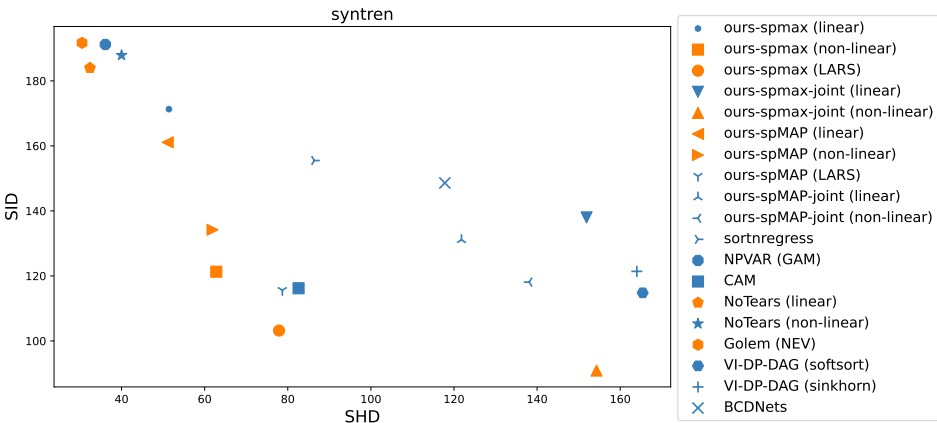

Figure 6: SHD vs SID on *SynTReN*. The solutions lying on the Pareto front are colored in orange and the others in blue.

**Sparsemax vs sparseMAP comparison** In Figures 8 and 7, we report an analysis of the effect of DAGuerreotype 's hyper-parameter $K$ and of the sample size $n$ on the quality of the learned DAG. Recall that $K$ corresponds to the maximal number of selected permutations for **sparsemax** and to the maximal number of iterations of the active set algorithm for **sparseMAP**, and that is the principal parameter that controls the computational cost of the ordering learning step in our framework. This analysis is carried out on data generated by a linear SEM from a scale-free graph with equal variance Gaussian noise. We choose this simple setting for two reasons: the true DAG can be identified from (enough) observational data only (Proposition 7.5, Peters et al., 2017); provided with the true topological ordering, the LARS estimator can identify the true edges. In this setting, we can then assess the quality of sparsemax and sparseMAP independently from the quality of the estimator. For reference, in Figures 8 and 7 we also report the performance of optimizing LARS with a random ordering or with a true one.

We observe that sparsemax and sparseMAP provide MAP orderings that are significantly better than random ones for $K > 2$ and for any sample size. For $K$ big enough, these orderings give DAGs that are almost as good as when knowing the variable ordering. Furthermore, apart when $K = 2$, increasing the sample size generally results in an improvement (although moderate) in the performance of sparsemax and sparseMAP. However, the gap from true's performance does not reduces when increasing $n$ or $K$, which can be explained by the non-convexity of the search space and DAGuerreotype getting stuck in local minima. We further observe that sparsemax generally provides better solutions than sparseMAP's at a comparable training time. The only settings where this is not the case are for $d = 30$ and $K > 35$. Notice that sparseMAP's performance peaks at $K = 35$ and degrades for higher $K$ in this setting. This phenomenon can be due to the inclusion of unnecessary orderings to sparseMAP's set, which ends up hurting training.

We further study in Figures 9 and 10 the effect of the pruning strength, (controlled by $\lambda$) on the performance of both operators on the real datasets. For this experiment, we instantiate DAGuerreotype with the linear estimator and train it by joint optimization. As a general remark, when strongly penalizing dense graphs (higher $\lambda$) SHD generally improves and SID degrades. On Sachs, the two operators do not provide significantly different results, while on SynTReN we find that sparsemax provides better SHD for comparable SID.

**Additional results on synthetic data** We report in Figures 11 and 12 an additional comparison of DAGuerreotype with several state-of-the-art baselines on synthetic problems of varying number of nodes $d$ and $n = 1,000$ samples generated from scale-free DAGs with $2d$ expected number of edges and different noise models: (*Gaussian*) linear SEM with equal variance Gaussian noise; (*Gumbel*) linear SEM with equal variance Gumbel noise; (*MLP*) 2-layer neural network SEM with sigmoid activations and equal variance Gaussian noise. To limit the varsortability of the generated problems, the parameters of the SEMs are uniformly drawn from $[-0.5, -0.1] \cup [0.1, 0.5]$. The resulting problems are still varsortable on average, as demostrated by the great performance of sortnregress, and by the fact that by initializing DAGuerreotype 's parameters $\boldsymbol{\theta}$ with the marginal variances of

Table 1: *Sachs*. We report Structural Hamming Distance (SHD, the lower the better), Structural Interventional Distance (SID, the lower the better), (F1, the higher the better), and the number of predicted edges for all methods.

| Method | SHD ↓ | SID ↓ | F1 ↑ | # edges |
|---|---|---|---|---|
| NoTears (linear) | 16.0 | 52.0 | 0.095 | 4 |
| NoTears (non-linear) | 14.0 | 50.0 | 0.320 | 8 |
| Golem (NEV) | 15.0 | 51.0 | 0.190 | 4 |
| VI-DP-DAG (softsort) | 43.0 | 37.0 | 0.265 | 51 |
| VI-DP-DAG (sinkhorn) | 42.0 | 38.0 | 0.269 | 50 |
| BCDNets | 39.0 | 46.0 | 0.196 | 34 |
| sortnregress | 13.1 | 50.8 | 0.377 | 9.0 |
| CAM | 30.0 | 47.0 | 0.28 | 33.0 |
| NPVAR (GAM) | 13.2 | 51.2 | 0.344 | 9.6 |
| DAGuerreotype -spmax (linear) | 13.8 | 51.6 | 0.323 | 7.7 |
| DAGuerreotype -spMAP (linear) | 13.8 | 49.1 | 0.309 | 7.6 |
| DAGuerreotype -spmax (LARS) | 13.8 | 51.7 | 0.316 | 9.5 |
| DAGuerreotype -spMAP (LARS) | 14.4 | 49.6 | 0.275 | 9.2 |
| DAGuerreotype -spmax (non-linear) | 13.5 | 50.9 | 0.348 | 7.1 |
| DAGuerreotype -spMAP (non-linear) | 14.1 | 48.1 | 0.32 | 7.4 |
| DAGuerreotype -spmax-joint (linear) | 16.9 | 44.9 | 0.36 | 16.1 |
| DAGuerreotype -spMAP-joint (linear) | 14.1 | 51.5 | 0.244 | 6.8 |
| DAGuerreotype -spmax-joint (non-linear) | 14.7 | 50.7 | 0.265 | 8.6 |
| DAGuerreotype -spMAP-joint (non-linear) | 14.4 | 51.1 | 0.236 | 5.0 |

Table 2: *SynTReN*. We report Structural Hamming Distance (SHD, the lower the better), Structural Interventional Distance (SID, the lower the better), Topological Ordering Pearson Correlation (TOPC, the higher the better), (F1, the higher the better) number of predicted edges all averaged over the 10 networks.

| Method | SHD ↓ | SID ↓ | F1 ↑ | # edges |
|---|---|---|---|---|
| NoTears (linear) | 32.4 | 184.0 | 0.157 | 17.7 |
| NoTears (non-linear) | 40.0 | 187.9 | 0.165 | 28.1 |
| Golem (NEV) | 30.5 | 191.7 | 0.152 | 15.4 |
| VI-DP-DAG (softsort) | 165.4 | 114.8 | 0.109 | 175.7 |
| VI-DP-DAG (sinkhorn) | 164.0 | 121.4 | 0.104 | 173.6 |
| BCDNets | 117.8 | 148.6 | 0.113 | 119.0 |
| sortnregress | 86.4 | 156.0 | 0.151 | 89.8 |
| CAM | 82.6 | 116.0 | 0.192 | 86.6 |
| NPVAR (GAM) | 36.1 | 191.0 | 0.184 | 26.0 |
| DAGuerreotype -spmax (linear) | 51.4 | 171.0 | 0.182 | 44.5 |
| DAGuerreotype -spMAP (linear) | 51.1 | 161.0 | 0.211 | 47.2 |
| DAGuerreotype -spmax (LARS) | 77.9 | 103.0 | 0.238 | 86.0 |
| DAGuerreotype -spMAP (LARS) | 78.7 | 116.0 | 0.212 | 83.8 |
| DAGuerreotype -spmax (non-linear) | 62.8 | 121.0 | 0.245 | 65.1 |
| DAGuerreotype -spMAP (non-linear) | 61.9 | 134.0 | 0.255 | 65.5 |
| DAGuerreotype -spmax-joint (linear) | 152.0 | 138.0 | 0.104 | 161.0 |
| DAGuerreotype -spMAP-joint (linear) | 122.0 | 131.0 | 0.139 | 131.0 |
| DAGuerreotype -spmax-joint (non-linear) | 154.0 | 90.9 | 0.149 | 169.0 |
| DAGuerreotype -spMAP-joint (non-linear) | 138.0 | 118.0 | 0.143 | 150.0 |

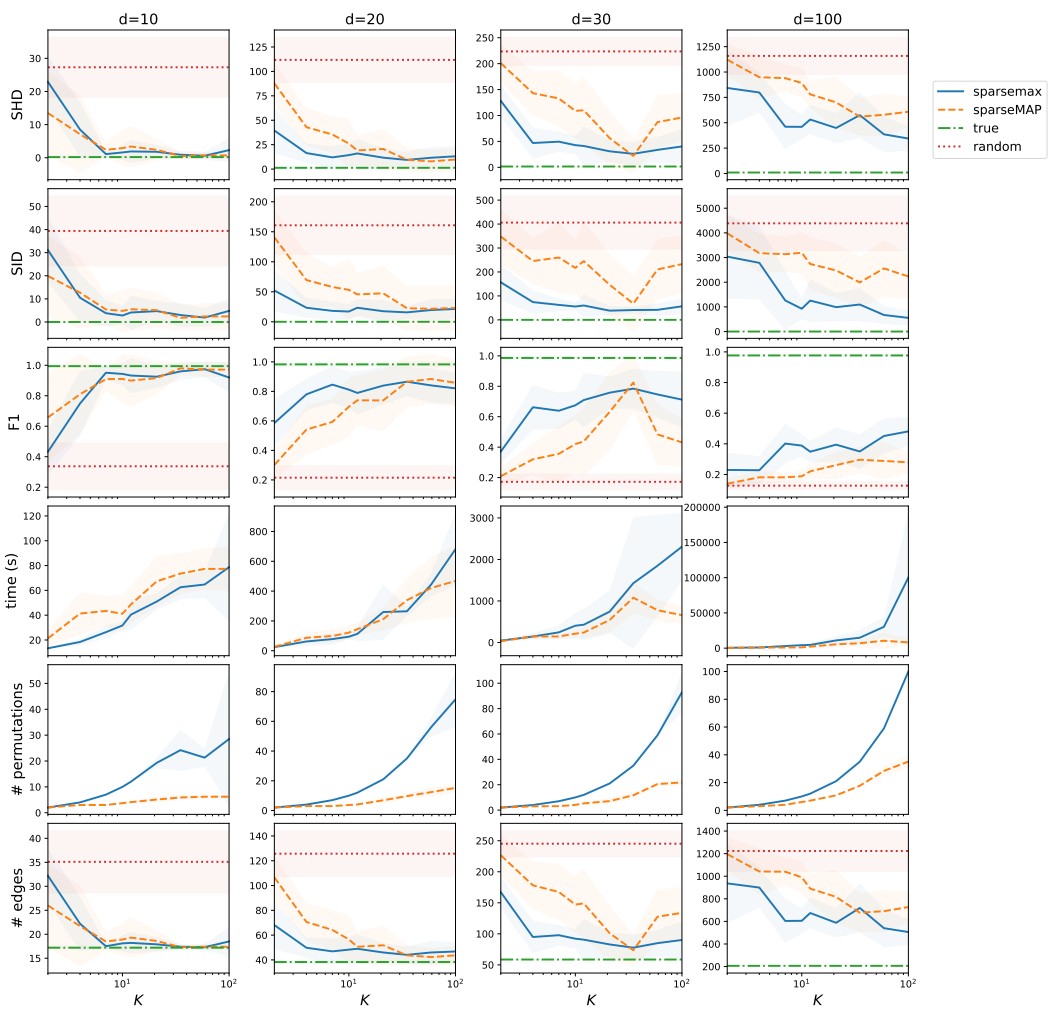

Figure 7: Comparison of different strategies for learning topological orderings, on data $(1,000$ samples) generated from a linear SEM with Scale-Free graph and Gaussian noise, and a varying number of nodes $d$. In order from top to bottom, we plot SHD, SID, F1, training time in seconds, number of permutations, and number of predicted edges (all at the end of training) as a function of the sparse operators' parameter $K$, which corresponds to the maximal number of sampled permutations for **sparsemax** and to the maximal number of iterations of the active set algorithm for **sparseMAP**. We also include two simple variants of DAGuerreotype, where the ordering is fixed to one true ordering (**true**) or to a random one (**random**). Results are averaged over 10 seeds.

the nodes consistently improves its performance, compared to initializing them with the vector of all zeros. For these experiments we set $K = 10$, use the linear edge estimator and jointly optimize all DAGuerreotype 's parameters.

Compared to other differentiable order-based methods, DAGuerreotype consistently provides a significantly better trade-off between SHD and SID, confirming our findings on the real-world data. Indeed, these baselines generally discover DAGs with high false positive rates. DAGuerreotype equipped with the sparseMAP operator also improves upon the linear continuous methods based on the exponential matrix regularization, but when equipped with the top-$k$ sparsemax operator its results on these settings depend on a good initialization of $\boldsymbol{\theta}$ (the marginal variances in this case) and worsen with the number of nodes. A higher value of $k$ would be required to improve DAGuerreotype -sparsemax's performance in these settings, as shown in Figure 8.

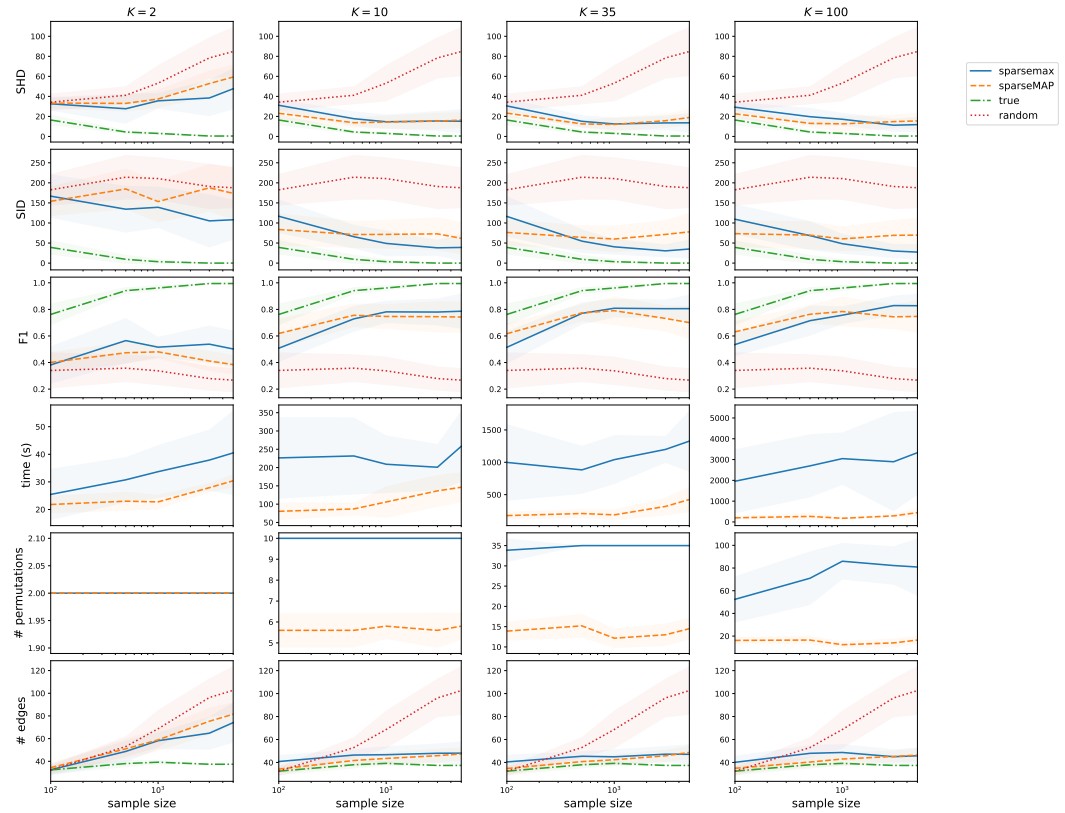

Figure 8: Comparison of different strategies for learning topological orderings, on samples of varying size ($n \in [100, 5'000]$ on the $x$-axes) generated from a linear SEM with Scale-Free graph and Gaussian noise, and number of nodes $d = 20$. In order from top to bottom, we plot SHD, SID, F1, training time in seconds, number of permutations, and number of predicted edges (all at the end of training) for 4 values of the sparse operators' parameter $K$, which corresponds to the maximal number of sampled permutations for **sparsemax** and to the maximal number of iterations of the active set algorithm for **sparseMAP**. We also include two simple variants of DAGuerreotype, where the ordering is fixed to one true ordering (**true**) or to a random one (**random**). Results are averaged over 10 seeds.

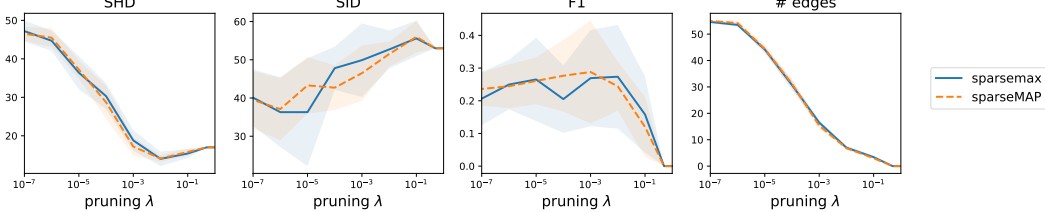

Figure 9: *Sachs*. Effect of L0 pruning intensity (controlled by $\lambda$) on SHD, SID, F1 and number of learned edges for DAGuerreotype jointly optimizing a linear estimator and the topological ordering distribution either with sparsemax (**sparsemax**) or sparseMAP (**sparseMAP**).

In terms of running times, DAGuerreotype is aligned with NoTears and is generally faster than CAM, Golem and VI-DP-DAG. Of course DAGuerreotype's running times strongly depend on the value of $K$, the choice of edge estimator and the optimization of either the joint or bi-level problems.

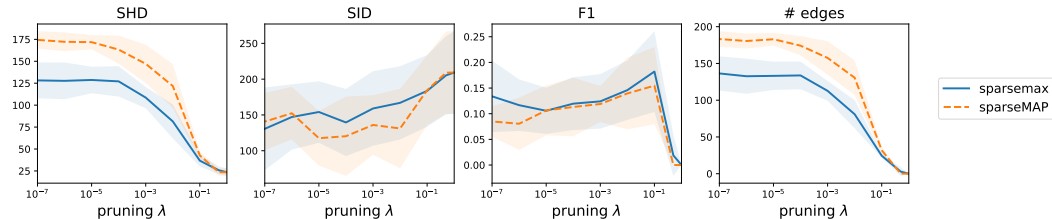

Figure 10: *SynTReN*. Effect of L0 pruning intensity (controlled by $\lambda$) on SHD, SID, F1 and number of learned edges for DAGuerreotype jointly optimizing a linear estimator and the topological ordering distribution either with sparsemax (**sparsemax**) or sparseMAP (**sparseMAP**).

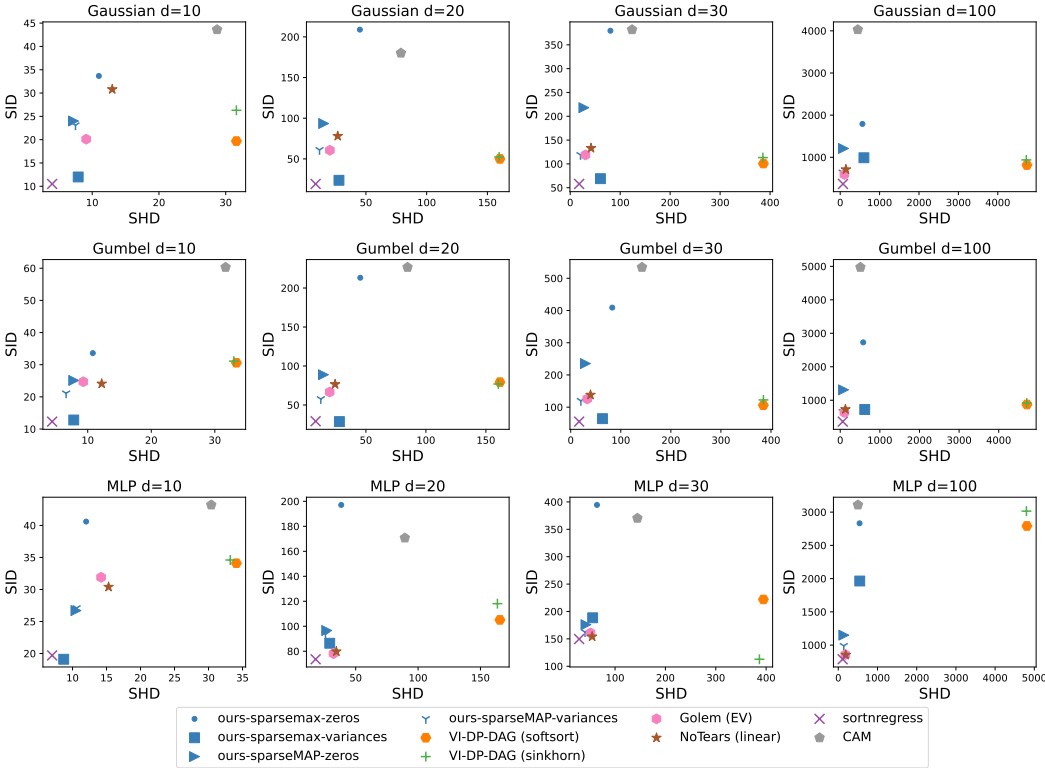

Figure 11: SHD vs SID on synthetic datasets generated from scale-free DAGs with Gaussian (top), Gumbel (middle) and MLP (bottom) SEMs. DAGuerreotype's (ours) $\theta$ is either initialized with a zero vector (zeros) or with the marginal variances (variances).

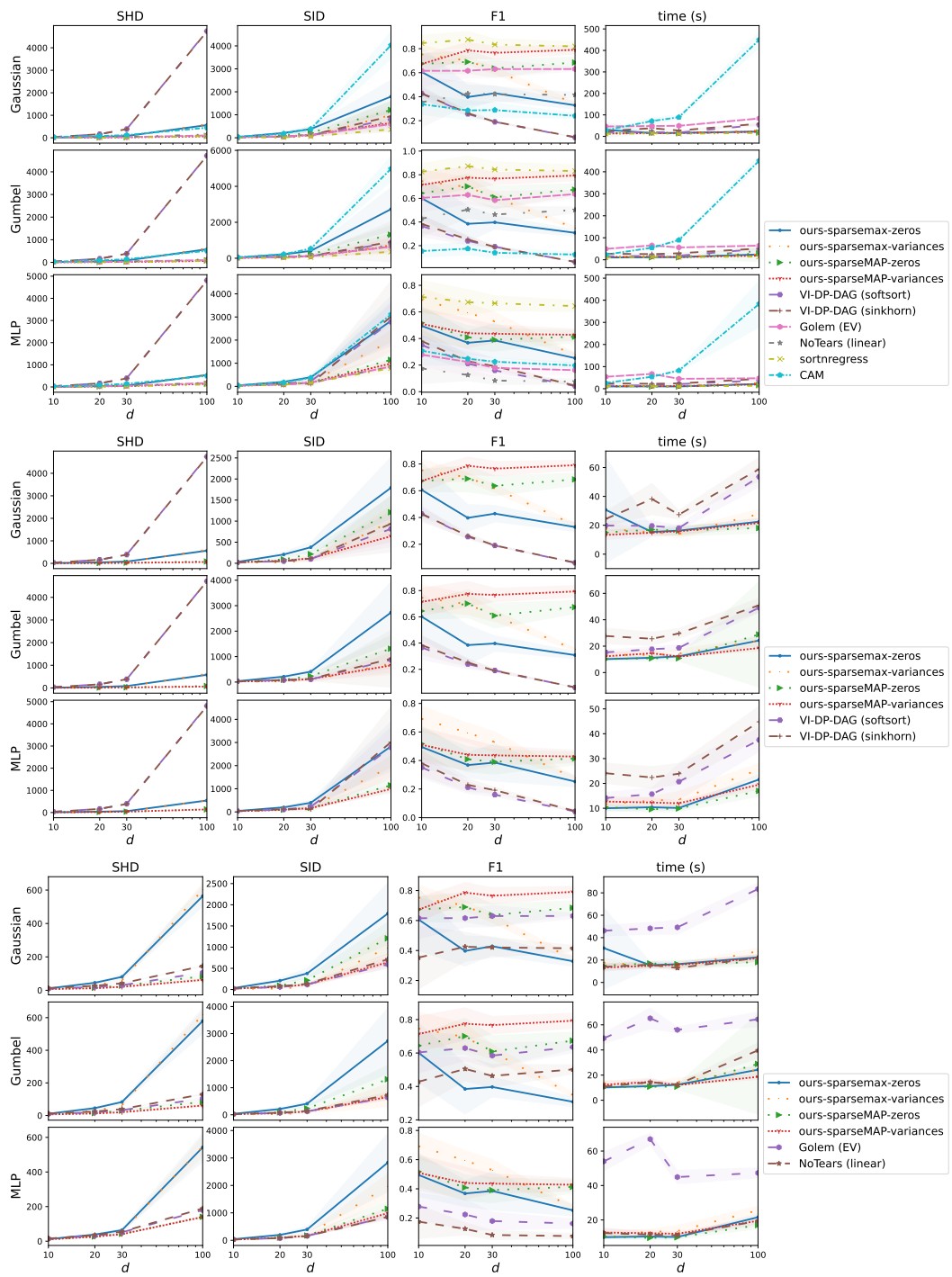

Figure 12: Comparison of DAGuerreotype (ours) with related methods in terms of SHD, SID, F1, training time (in seconds) on synthetic datasets generated from scale-free DAGs with Gaussian, Gumbel and MLP SEMs. DAGuerreotype's (ours) $\theta$ is either initialized with a zero vector (zeros) or with the marginal variances (variances). For ease of reading, we split the full comparison (top) into two, to focus on the comparison with differentiable order-based methods (middle) and with differentiable methods based on the matrix exponential constraint (bottom).

