# OpenReview forum: "DAG Learning on the Permutahedron"
_ICLR.cc/2023/Conference — ICLR 2023 poster_

### Official Review · Reviewer_SU9Y · 2022-10-24

**Confidence:** 2
**Correctness:** 4
**Technical Novelty And Significance:** 3
**Empirical Novelty And Significance:** 3
**Recommendation:** 6

**Clarity, Quality, Novelty And Reproducibility:**

The paper is very well written, the description of the related work is adequate, and the contributions are clear.
As far as I know, the proposed method is novel, producing good results.

The code is included as supplementary material, which is much appreciated for reproducibility.

There seems to be a minor typo in equation (3): the r in the simplex.


**Strength And Weaknesses:**

The strengths of the paper are the clarity, the observation of the optimization over the permutahedron, and the fact that it is actually a framework where one can choose different methods for the sub-tasks.

**Summary Of The Paper:**

In this paper, the authors propose a framework for DAG learning from data. The method consists of a two-step procedure, where the first one is based on an observation about optimizing the order of nodes over the permutahedron. Then the edges to keep are selected by using a regularization over the set of functions.
The method is tested on two real-world datasets with promising results.


**Summary Of The Review:**

The authors of the paper propose an interesting two-step framework for DAG learning, where different methods can be used for the sub-tasks.
It is well written, and the results are promising.

---

> ### Author Response · Authors · 2022-11-14
> **thank you for the positive evaluation**
>
> We warmly thank the reviewer for their time and for their positive comments. Many thanks also for spotting the typo in equation (3) which we corrected in the new version. We appreciate the acknowledgement of writing clarity and code release for reproducibility.
>
> We are at the reviewer’s disposal if there are any other questions or suggestions on how we can further strengthen our submission.
>
> Thank you again.

---

### Official Review · Reviewer_kPDZ · 2022-10-25

**Confidence:** 3
**Correctness:** 3
**Technical Novelty And Significance:** 2
**Empirical Novelty And Significance:** 2
**Recommendation:** 6

**Clarity, Quality, Novelty And Reproducibility:**

The paper is generally clear. The proposed method may be rather incremental compared to existing ones. See above comments for more details.

**Strength And Weaknesses:**

Strength:
- A new continuous approach to traverse the DAG space is provided. Solving continuous DAG learning task is a topic with significance.
- The approach is flexible and accommodates non-differentiable black-box estimators.

Weakness:
- As the paper mentioned, the possibility of using differentiable approach to learn DAGs via ordering has been proposed by (Cundy et al., 2021; Charpentier et al., 2022). Therefore, the proposed method may seem incremental, i.e., optimizing over permutation vectors instead of permutation matrices as in existing works.
    - I would suggest the authors to provide a detailed discussion of the advantages of their method over (Cundy et al., 2021; Charpentier et al., 2022). Also, can those existing methods be modified to accommodate non-differentiable black-box estimators?
- The paper argued that existing works (Cundy et al., 2021; Charpentier et al., 2022) required expensive optimization over the permutation matrices. However, the paper currently lacks empirical studies to support it--either comparison of running time or scalability to very large graphs would be more compelling to demonstrate the efficiency/scalability of the proposed method.
-  Although the paper mentioned a recent study by Reisach et al. (2021) regarding the flaw of synthetic dataset in DAG learning benchmark, I would still suggest the authors to compare their method to the baselines on those synthetic benchmarks to have an idea about how the method performs. This is because for real world datasets, the model assumption (e.g.,. linear or nonlinear additive noise models) may not hold.
- In 'over-relaxation' part of Section 1, several methods based on penalizing the 'DAG-ness' of the adjacency matrix are actually *guaranteed to return a valid DAG* under some conditions, as shown by the following papers. Therefore the statement is not entirely true and should be corrected.
    - Remark 2 of Ng et al. (2022). On the Convergence of Continuous Constrained Optimization for Structure Learning.
    - Lemma 6 of Bello et al. (2022). DAGMA: Learning DAGs via M-matrices and a Log-Determinant Acyclicity Characterization.

----------------
I appreciate the detailed response from the authors, which has addressed my concern. I have increased my score accordingly.

**Summary Of The Paper:**

The paper develops a continuous optimization method for learning DAGs by optimizing over the polytope of permutation vectors. Two versions of the methods are provided: (1) alternately iterates between node-ordering and edge-optimization, and (2) optimizes them jointly. The resulting method has several advantages over existing methods, such as (1) returning exact DAGs and (2) accommodating non-differentiable black-box estimators. Experiment results on two real-world datasets are provided.

**Summary Of The Review:**

- The proposed method may be rather incremental compared to existing ones.
- Lack of experiment results on synthetic benchmarks.

---

> ### Author Response · Authors · 2022-11-14
> **new experiments and clarification**
>
> We thank the reviewer for their time and for the detailed and thoughtful feedback. In the updated version of the paper we reworded the contribution list in Sec. 1 to make the novelty of our approach clearer. We also edited Sections 3 and 4 to better separate background to original contributions and further better link key results that in the previous version were only presented in the appendix, specifically regarding global sensitivity analysis of our vector parameterization (Theorem 1) and the development of the top-k permutations oracle (Algorithm 1, with correctness proof in Appendix A). We address all comments below.
>
> [..provide a detailed discussion of the advantages … over (Cundy et al., 2021; Charpentier et al., 2022)..]
>
> Thank you for bringing this up! We agree it would improve the paper to include this discussion. The advantages of our method over (Cundy et al., 2021; Charpentier et al., 2022) are: (a) our parametrization, based on sorting, improves efficiency in practice (Fig. 11) and has theoretically stabler learning dynamics as measured by our bound on SHD (Theorem 1); (b) our method allows for any downstream edge estimator, including non-differentiable ones, critically allowing for black box estimators; (c) empirically our method vastly improves over both approaches in terms of SID and especially SHD on both real-world and synthetic data. Thank you again for mentioning this, we have included this in our updated version.
>
> [... optimizing over permutation vectors instead of permutation matrices…incremental…]
>
> We’ve updated the paper to clarify this: our d-dimensional parametrization on the permutahedron is a radically different perspective compared to the existing d^2 parametrizations (based on permutation matrices), which leads to several advantages. Our parametrization, based on sorting, improves efficiency in practice (Fig. 11) and has theoretically stabler learning dynamics as measured by our bound on SHD (Theorem 1).
>
> Furthermore, our novel bilevel formulation allows learning with arbitrary downstream estimators, including off-the-shelf non-differentiable ones. Implementing our strategy also required deriving a novel algorithm for computing the top-k scoring permutations (Algorithm 1, with details in Appendix A), using insights from combinatorial optimization.
>
> [... comparison of running time … compare their method to the baselines on those synthetic benchmarks … ]
>
> Following this suggestion, we extended our set of experiments on synthetic data of Appendix D to include a comparison with the baselines in terms of SHD, SID, F1 and runtimes (Figures 10 and 11). Compared to differentiable order-based methods, ours is better at discovering the graph and is generally faster in practice, especially on larger graphs. Compared to methods based on the trace of the exponential matrix, our approach returns better graphs when equipped with the sparseMAP operator on these settings. Training times are generally comparable with those of NoTears and lower than Golem.
>
> [...several methods based on penalizing the 'DAG-ness' of the adjacency matrix are actually guaranteed to return a valid DAG…]
>
> We thank the reviewer for pointing us to these recent works analyzing the convergence of methods based on the matrix exponential acyclicity constraint and validity of the found solution. We have included them and rephrased our statement in the revision.
> More precisely, we clarified that existing methods based on powers of the adjacency matrix for acyclicity and the Augmented Langrangian method for optimization (e.g., NoTears) are not guaranteed to converge to a DAG (as proved in [Ng et al. 2022]). This would be the case if they were solved by the Quadratic Penalty method, but only when penalization weight tends to infinity, which practical implementations avoid.
> We were unaware of the concurrent work [Bello et al. 2022] which proposes a new characterization of acyclicity and a new optimization problem which is guaranteed to converge to a DAG. We have clarified that also for this approach there is no guarantee during training that the optimized graph is always a DAG. Thank you again for pointing these out!

---

### Official Review · Reviewer_EZMX · 2022-10-25

**Confidence:** 3
**Correctness:** 4
**Technical Novelty And Significance:** 3
**Empirical Novelty And Significance:** 3
**Recommendation:** 6

**Clarity, Quality, Novelty And Reproducibility:**

The paper is well-written and clear. The algorithm has a combination of important ideas. The experiments look reproducible to me.

**Strength And Weaknesses:**

Strength: the proposed solution is flexible and more efficient and more accurate than the existing methods. Comparison with the related work has been covered in great detail. The experimental evidence is extensive, consisting of the best algorithms known before and includes both benchmark and synthetic datasets.

Weakness: Ultimately, the paper reads like a conglomeration of existing ideas. I could not clearly find a single new technique that the authors have found.

**Summary Of The Paper:**

In this paper, the authors look at the problem of recovering the DAG structure from samples from a DAG-structured distribution. Despite having seen a lot of important works spanning the last several decades, the problem has remained open and is an active area of research. The proposed solution in this paper consists of two steps i) finding a topological ordering of the DAG, ii) discovering edges according to the ordering. The authors manage to do both steps which is end-to-end on the one hand and accomodates any black-box edge discovery algorithm for step ii).

Experiments conducted by the authors reveal that their algorithm are better in terms of both the metrics SHD and SID. In fact, their solutions learn at the Pareto frontier of these two metrics.

**Summary Of The Review:**

Given the superior experimental performance of the proposed approach, I tend to accept the paper.

---

> ### Author Response · Authors · 2022-11-14
> **contributions and novelty**
>
> We warmly thank the reviewer for their time and thoughtful comments, in particular for highlighting the advantages of our method and the quality of our analysis!
>
> [...conglomeration of existing ideas…]
>
> We have updated the paper to make the novelty of our approach clearer, rewording the contribution list in Sec. 1 and revising Sec 3 and 4 to better separate background to original contributions. In the initial submission, some of our new results were in the appendix and not properly highlighted in the main paper. Specifically we have made the following changes:
>
> 1. We clarify that our d-dimensional parametrization on the permutahedron is a radically different perspective compared to the existing d^2 parametrizations, which leads to several advantages. Our parametrization, based on sorting, improves efficiency in practice (Fig. 11) and has theoretically stabler learning dynamics as measured by our bound on SHD (Theorem 1).
> 2. Our novel bilevel formulation allows learning with arbitrary downstream estimators, including, as you identified, off-the-shelf non-differentiable ones.
> 3. Implementing our strategy required deriving a secondary novel algorithm: computing the top-k scoring permutations (Algorithm 1, with details in Appendix A) using combinatorial optimization insights (the algorithm is used by our best performing variants in Figure 2).

---

### Official Review · Reviewer_bSZG · 2022-10-29

**Confidence:** 4
**Correctness:** 3
**Technical Novelty And Significance:** 3
**Empirical Novelty And Significance:** 2
**Recommendation:** 6

**Clarity, Quality, Novelty And Reproducibility:**

Clarity: Most of the paper is clear, but a few wordings for me were confusing. For instance, the term over-relaxation in the introduction is somewhat unclear and to me does not reflect what is said afterwards. In fact, I should bring to the authors' attention a recent work [1] that studies some disadvantages of existing acyclicity regularizers more formally, and perhaps consider referencing it for the next revision. In Section 2, it is stated "the constraint on acyclicity is expressed as a smooth function and then relaxed to allow efficient optimization", what do the authors mean by "relaxed"?

Quality and Novelty: The work in general is of good quality, and although it does not shine in technical contributions, I think the paper nicely leverages work on sparse relaxations for the problem of DAG learning.

Reproducibility: Authors have shared code for reproducibility so that's a plus, although I have not tested it myself.


[1] Bello et al. (2022) “DAGMA: Learning DAGs via M-matrices and a Log-Determinant Acyclicity Characterization”. NeurIPS.


**Strength And Weaknesses:**

Strengths: I generally find the approach interesting and a nice mix of ideas from the sparse relaxation literature to the DAG learning problem. The paper is also generally well-written modulo some unspecified notation.

Weaknesses: The main weakness is the experimental section. Given that this is not a theoretical work, one should at least expect a more comprehensive set of experiments. The work of Reisach et al. (2021) is cited for skipping comprehensive synthetic experiments basically, and both real-world networks experimented on have a small number of nodes. The scaling problem when using OLS has been known for a while for linear regression, and hence the same is expected to happen for linear SEMs, this is why methods such as PC are robust to this problem since PC works by performing CI tests---Reisach et al. (2021). To alleviate the marginal variance problem, one can just use a bit smaller weights for the simulations. It would be great to see how the proposed method behaves against other baselines for ER, SF graphs for different numbers of nodes. Moreover, it is a must to compare the runtimes as well, given that other methods such as NOTEARS do not have a clear computational complexity, it is important to see what the actual performance is in terms of runtime.

From the experiments, I would like to conclude: Is the proposed method better? Is it faster? Both? I cannot conclude that from the current set of experiments.

**Summary Of The Paper:**

The paper studies the problem of learning DAGs from observational data. Different from prior work under the continuous framework, the authors propose an approach to optimize over the polytope of permutations, where edges can be jointly or conditionally optimized given a particular topological ordering. The key to the approach is to leverage recent progress on sparse relaxation methods such as top-k sparsemax and sparseMAP. Some experiments on real data are provided.

**Summary Of The Review:**

My main concerns are on about the experiments, it is not clear if the proposed approach works well for linear/nonlinear models, for different dimensions, and for different types of graphs.

---

> ### Author Response · Authors · 2022-11-14
> **new experiments, clarification**
>
> We warmly thank the reviewer for their time and for the constructive feedback. Many thanks also for pointing us to the recent analysis of the exponential matrix acyclicity constraint [1], which we have included in the related work. We address all comments below.
>
> [..how the proposed method behaves against other baselines for ER, SF graphs..compare runtimes..]
>
> Following this suggestion, we extended our set of experiments on synthetic data of Appendix D to include a comparison with the baselines in terms of SHD, SID, F1 and runtimes (Figures 10 and 11). Our approach outperforms existing differentiable order-based methods on these graph prediction metrics, and is generally faster in practice. Compared to methods based on the trace of the exponential matrix, our approach returns better graphs when equipped with the sparseMAP operator on these settings. Training times are generally comparable with those of NoTears and lower than Golem.
>
> [..over-relaxation/relaxed..]
>
> Thank you for pointing out this wording, we agree this could be clearer. By ‘relaxation’ we meant that the way the acyclicity constraint is enforced (in the Augmented Lagrangian method) does not guarantee acyclicity during training and most importantly at convergence. We have updated the text to clarify this point.
>
> [...technical contribution…]
>
> We realized that a few key results in the appendix were not properly highlighted in the main paper. Therefore we have migrated these to Section 4 to better highlight our new contributions: (a) global sensitivity analysis of our vector parameterization (Theorem 1) and, (b) the development of the top-k permutations oracle (Algorithm 1, with full correctness proof in Appendix A). Additionally we reworded the contribution description in Section 1 to more clearly represent our original contributions.

---

### Author Response · Authors · 2022-12-02
**Discussion**

Dear Reviewers,

Thank you for your time and reviews. Our point-by-point responses can be found below. We have revised our paper based on your suggestions and comments.

As it is approaching the end of the Author-Reviewer Discussion, could you take a look at our response and revision, and let us know if we have addressed all your concerns? We are happy to continue discussions if you still have remaining concerns.

Thank you for your time!

-- The Authors

---

### Decision · Program_Chairs · 2023-01-20

**Decision:**

Accept: poster

**Justification For Why Not Higher Score:**

Not a fundamental contribution.

**Justification For Why Not Lower Score:**

The paper could be of interest to the audience given importance of learning DAGs.

**Metareview: Summary, Strengths And Weaknesses:**

The paper is in the class of work that suggest using continuous optimisation approaches for DAG learning. The main idea/contribution is leveraging recent approaches in sparse relaxation methods (e.g.,  top-k sparsemax and sparseMAP) for DAG learning. I do not feel that the technical contribution is adequately novel but given the importance of DAG learning in causal inference and Bayesian inference, the paper could be of interest to the learning community.


**Note From Pc:**

if the above contains the word "oral" or "spotlight" please see: "oral" presentation means -> notable-top-5% and "spotlight" means -> notable-top-25%. As stated in our emails, we are disassociating presentation type from AC recommendations